# Outliers with Opposing Signals Have an Outsized Effect on Neural Network Optimization

**Elan Rosenfeld**
Carnegie Mellon University
elan@cmu.edu

**Andrej Risteski**
Carnegie Mellon University
aristesk@andrew.cmu.edu

## Abstract

We identify a new phenomenon in neural network optimization which arises from the interaction of depth and a particular heavy-tailed structure in natural data. Our result offers intuitive explanations for several previously reported observations about network training dynamics, including a conceptually new cause for progressive sharpening and the edge of stability. We further draw connections to related phenomena including grokking and simplicity bias. Experimentally, we demonstrate the significant influence of paired groups of outliers in the training data with strong *Opposing Signals*: consistent, large magnitude features which dominate the network output and provide gradients which point in opposite directions. Due to these outliers, early optimization enters a narrow valley which carefully balances the opposing groups; subsequent sharpening causes their loss to rise rapidly, oscillating between high on one group and then the other, until the overall loss spikes. We carefully study these groups' effect on the network's optimization and behavior, and we complement this with a theoretical analysis of a two-layer linear network under a simplified model. Our finding enables new qualitative predictions of training behavior which we confirm experimentally. It also provides a new lens through which to study and improve modern training practices for stochastic optimization, which we highlight via a case study of Adam versus SGD.

## 1 Introduction

There is a steadily growing list of intriguing properties of neural network (NN) optimization which are not readily explained by prior tools from optimization. Likewise, there exist varying degrees of understanding of the mechanistic causes for each. Extensive efforts have led to possible explanations for the effectiveness of Adam (Kingma & Ba, 2014), Batch Normalization (Ioffe & Szegedy, 2015) and other tools for successful training—but the evidence is not always entirely convincing, and there is certainly little theoretical understanding. Other findings, such as grokking (Power et al., 2022) or the edge of stability (Cohen et al., 2021), do not have immediate practical implications, but they provide new ways to study what sets NN optimization apart. These phenomena are typically considered in isolation—though they are not completely disparate, it is unknown what specific underlying factors they may share. Clearly, a better understanding of NN training dynamics in a specific context can lead to algorithmic improvements (Chen et al., 2021); this suggests that any commonality will be a valuable tool for further investigation.

In this work, we identify a phenomenon in NN optimization which offers a new perspective on many of these prior observations and which we hope will contribute to a deeper understanding of how they may be connected. While we do not (and do not claim to) offer a complete explanation for this finding, we present strong qualitative and quantitative evidence which suggests a more coherent picture of the origin of these prior results, offering a single high-level idea which naturally fits into several existing narratives. Specifically, we demonstrate the prevalence of paired groups of outliers which have a significant influence on a network's optimization dynamics. These groups are characterized by the inclusion of one or more (relatively) large magnitude features that dominate the network's output at random initialization and throughout most of training. In addition to their magnitude, the other distinctive property of these features is that they provide large, consistent, and *opposing* gradients, in that reducing loss on one group (usually) increases loss on the other; because of this structure, we refer to them as *Opposing Signals*. These features share a non-trivial correlation

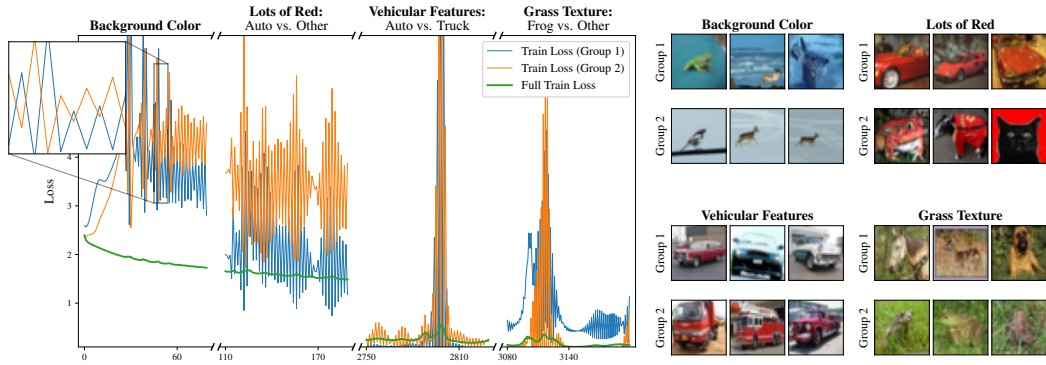

Figure 1: **Training dynamics of neural networks are heavily influenced by outliers with opposing signals.** We plot the overall loss of a ResNet-18 trained with GD on CIFAR-10, plus the losses of a small but representative subset of outlier groups. These groups have consistent *opposing signals* (e.g., wheels and headlights sometimes means car and sometimes truck). Throughout training, losses on these groups oscillate with growing amplitude—this oscillation has an obvious correspondence to the short term spikes in overall training loss and, in particular, appear to be the direct cause of the "edge-of-stability" phenomenon.

with the target task, but they are often not the "correct" (e.g. human-aligned) signal. In fact, in many cases these features perfectly encapsulate the classic conundrum of "correlation vs. causation"—for example, a bright blue sky background does not determine the label of a CIFAR image, but it does most often occur in images of planes. Other features *are* relevant, such as the presence of wheels and headlights on trucks and cars, or whether punctuation comes before after the end of a quotation.

Opposing signals are most easily understood with an example, which we will give along with a brief outline of their effect on training dynamics; a more detailed description is presented in Section 3. Fig. 1 depicts the training loss of a ResNet-18 trained with full-batch gradient descent (GD) on CIFAR-10, along with a few dominant outlier groups and their respective losses. In the early stages of training, the network enters a narrow valley in weight space which carefully balances the pairs' opposing gradients; subsequent sharpening of the loss landscape (Jastrzębski et al., 2020; Cohen et al., 2021) causes the network to oscillate with growing magnitude along particular axes, upsetting this balance. Returning to our example of a sky background, one step results in the class plane being assigned greater probability for all images with sky, and the next will reverse that effect. In essence, the "sky = plane" subnetwork grows and shrinks.[1] The direct result of this oscillation is that the network's loss on images of planes with a sky background will alternate between sharply increasing and decreasing with growing amplitude, with the exact opposite occurring for images of *non*-planes with sky. Consequently, the gradients of these groups will alternate directions while growing in magnitude as well. As these pairs represent a small fraction of the data, this behavior is not immediately apparent from the overall training loss—but eventually, it progresses far enough that the overall loss spikes. As there is an obvious direct correspondence between these two events throughout, we conjecture that opposing signals are the direct cause of the "edge-of-stability" phenomenon (Cohen et al., 2021). We also note that the most influential signals appear to increase in complexity over time (Nakkiran et al., 2019).

We repeat this experiment across a range of vision architectures: though the precise groups and their order of appearance change, the pattern occurs consistently. We also verify this behavior for transformers on next-token prediction of natural text and small ReLU MLPs on simple 1D functions; we give some examples of opposing signals in text in Appendix B. However, we rely on images for exposition because it offers the clearest intuition. To isolate this effect, most of our experiments use GD—but we observe similar patterns during SGD, which we present in Section 4.

---

[1] It would be more precise to say "strengthening connections between regions of the network's output and neurons which have large activations for sky-colored inputs". Though we prefer to avoid informal terminology, this example makes clear that the more relaxed phrasing is usually much cleaner. We therefore employ it when the intended meaning is clear.

**Contributions.** The primary contribution of this paper is demonstrating the existence, pervasiveness, and large influence of opposing signals during NN optimization. We further present our current best understanding, with supporting experiments, of how these signals *cause* the observed training dynamics—in particular, we provide evidence that it is a consequence of depth and steepest descent methods. We complement this discussion with a toy example and an analysis of a two-layer linear net on a simple model. Notably, though rudimentary, our explanation enables concrete qualitative predictions of NN behavior during training, which we confirm experimentally. It also provides a new lens through which to study modern stochastic optimization methods, which we highlight via a case study of SGD vs. Adam. We see possible connections between opposing signals and a wide variety of phenomena in NN optimization and generalization, including *grokking* (Power et al., 2022), *catapulting/slingshotting* (Lewkowycz et al., 2020; Thilak et al., 2022), *simplicity bias* (Valle-Perez et al., 2019), *double descent* (Belkin et al., 2019; Nakkiran et al., 2020), and Sharpness-Aware Minimization (Foret et al., 2021). We discuss these and other connections in Appendix A.

## 2  SETUP AND EXPERIMENTAL METHODOLOGY

Though their influence on aggregate metrics is non-obvious, identifying opposing signals is straightforward. When training a network with GD, we track its loss on each individual training point. For a given iteration, we identify the training points whose loss exhibited the most positive and most negative change in the preceding step (there is large overlap between these sets in successive steps). This set will sometimes contain multiple opposing signals, which we distinguish via visual inspection. This last detail means that the images we depict are not random, but we emphasize that it would not be correct to describe this process as cherry-picking: though precise quantification is difficult, these signals consistently obey the maxim "I know it when I see it". To demonstrate this fact, Appendix H contains the pre-inspection samples for a ResNet-18, VGG-11, and a Vision Transformer at several training steps and for multiple seeds ; we believe the implied groupings are immediate, even if not totally objective. We see algorithmic approaches to automatically clustering these samples as a direction for future study—for example, one could select samples by correlation in their loss time-series, or by gradient alignment.

Given how these samples were selected, several other characterizations seem relevant. For instance, maximal one-step loss change is often a reasonable proxy for maximum gradient norm; we could also consider the largest eigenvalue of the loss Hessian of the *individual point*, or how much curvature it has in the direction of the overall loss's top eigenvector. For large networks these options are far more compute-intensive than our chosen method, but we can evaluate them post-hoc on specific groups. In Fig. 2 we track these metrics for several opposing group pairs and find that they are consistently much larger than that of random samples from the training set.

**On the Possibility of a Formal Definition.** Though the features and their exemplar samples are immediately recognizable, we do not attempt to *exactly* define a "feature", nor an "outlier" with respect to that feature. The presence of a particular feature is often ambiguous, and it is difficult

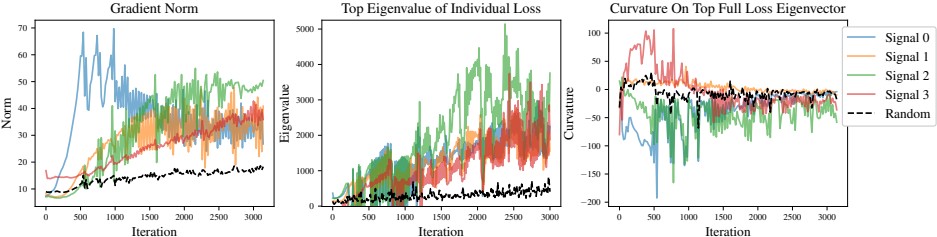

Figure 2: **Tracking other metrics which characterize outliers with opposing signals.** For computational considerations, we select outliers according to maximal per-step change in loss. However, this quantity relates to other useful metrics, such as gradient norm and curvature. We see that the samples we uncover are also significant outliers according to these metrics.

to define a clear threshold for what makes a given point an outlier.[2] Thus, instead of trying to exactly partition the data, we simply note that these heavy tails *exist* and we use the most obvious outliers as representatives for visualization. In Figs. 1 and 2 we choose an arbitrary cutoff of twenty samples per group. We also note that what qualifies as an opposing signal or outlier may vary over time. For visual clarity, Fig. 1 depicts the loss on only the most dominant group pair in its respective training phase, but this pattern occurs simultaneously for many different signals and at multiple scales throughout training. Further, the opposing signals are with respect to the model's internal representations (and the label), not the input space itself; this means that the definition is also a property of the architecture. For example, following Cohen et al. (2021) we train a small MLP to fit a Chebyshev polynomial on evenly spaced points in the interval $[-1, 1]$ (Fig. 36 in the Appendix). This data has no "outliers" in the traditional sense, and it is not immediately clear what opposing signals are present. Nevertheless, we observe the same alternating behavior: we find a pair where one group is a small interval of $x$-values and the opposing group contains its neighbors, all in the range $[-1, -0.5]$. This suggests that the network has internal activations which are heavily influential only for more negative $x$-values. In this context, these two groups are the outliers.

## 3 Understanding the Effect of Opposing Signals

Our eventual goal will be to derive actionable insights from this finding. To do this, it is necessary to gain a better understanding of *how* these opposing signals lead to the observed behavior. In this section we give a simplified "mental picture" which serves as our current understanding this process: first a general discussion of why opposing signals are so influential, followed by a more mechanistic description with a toy example. This explanation is intentionally high-level, but we will eventually see how it gives concrete predictions of specific behaviors, which we then verify on real networks. Finally, we prove that this behavior occurs on a two-layer linear network under a simple model.

### 3.1 Progressive Sharpening, and Why These Outliers are so Influential

At a high level, most variation in the input is unneeded when training a network to minimize predictive error—particularly with depth and high dimension, only a small fraction of information will be propagated to the last linear layer (Huh et al., 2021). Starting from random initialization, training a network aligns adjacent layers' singular values (Saxe et al., 2013; Mulayoff & Michaeli, 2020) to amplify meaningful signal while downweighting noise,[3] growing *sensitivity* to the important signal. This sensitivity can be measured, for example, by the spectral norm of the input-output Jacobian, which grows during training (Ma & Ying, 2021); it has also been connected to growth in the norm of the output layer (Wang et al., 2022). Observe that with this growth, small changes to *how the network processes inputs* become more influential. Hypothetically, a small weight perturbation could massively increase loss by redirecting unhelpful noise to the subspace to which the network is most sensitive, or by changing how the last layer uses it. The increase of this sensitivity thus represents precisely the growth of loss Hessian spectrum, with the strength of this effect increasing with depth (Wang et al., 2016; Du et al., 2018; Mulayoff & Michaeli, 2020).[4]

Crucially, this sharpening also depends on the structure of the input. If the noise is independent of the target, it will be downweighted throughout training. In contrast, *genuine signals which oppose each other* will be retained and perhaps even further amplified by gradient descent; this is because the "correct" feature may be much smaller in magnitude (or not yet learned), so using the large, "incorrect" feature is often the most immediate way of minimizing loss. As a concrete example, observe that a randomly initialized network will lack the features required for the subtle task of distinguishing birds from planes. But it *will* capture the presence of sky, which is very useful for reducing loss on such images by predicting the conditional $p(\text{class} \mid \text{sky})$ (this is akin to the "linear/shallow-first" behavior described by Nakkiran et al. (2019); Mangalam & Prabhu (2019)). Thus, any method attempting to minimize loss as fast as possible (e.g., steepest descent) may actually upweight these

---

[2]In the case of language—where tokenization is discrete and more interpretable—a precise definition is sometimes possible, such as those depicted in Appendix B.

[3]In this discussion we use the term "noise" informally. We refer not necessarily to pure randomness, but more generally to input variation which is least useful in predicting the target.

[4]The coincident growth of these two measures was previously noted by Ma & Ying (2021); Gamba et al. (2023); MacDonald et al. (2023), though they did not make explicit this connection to how the network processes different types of input variance.

Figure 3: **A toy example illustrating the effect of opposing signals.** Images with many blue pixels cause large activations and steep gradients. We project the loss to the hypothetical weight-space dimension "sky = plane". **Left:** Early optimization approaches the minimum, balancing the two opposing losses. Progress continues through this valley, further aligning subnetworks and growing the linear head. **Right**: The valley sharpens and iterates become unstable. Because these images are a small minority and the others are less sensitive to this axis, the train loss is not noticeably affected at first. Eventually either (a) the outlier gradients' growth forces the network to downweight "sky", flattening the valley and returning to the first phase; or (b) the weights "catapult" to a different basin.

features. Furthermore, amplified opposing signals will cause greater sharpening than random noise, because using a signal to the benefit of one group is maximally harmful for the other—e.g., confidently predicting plane whenever there is sky will cause enormous loss on images of other classes with sky. Since random noise is more diffuse, this effect is less pronounced.

This description is somewhat abstract. To gain a more precise understanding, we illustrate the dynamics more explicitly on a toy example.

## 3.2 Illustrating with a Hypothetical Example of Gradient Descent

Consider the global loss landscape of a neural network: this is the function which describes how the loss changes as we move through parameter space. Suppose we identify a direction in this space which corresponds to the network's use of the "sky" feature to predict plane versus some other class. That is, we will imagine that whenever the input image includes a bright blue background, moving the parameters in one direction increases the logit of the plane class and decreases the others, and vice-versa. We will also decompose this loss—**among images with a sky background, we consider _separately_ the loss on those labeled plane versus those with any other label.** Because the sky feature has large magnitude, a small change in weight space will induce a large change in the network outputs— i.e., a small movement in the direction "sky = plane" will greatly increase loss on these non-plane images.

Fig. 3 depicts this heavily simplified scenario. Early in training, optimizing this network with GD will rapidly move towards the minimum along this direction. In particular, until better features are learned, the direction of steepest descent will lead to a network which upweights the sky feature and predicts $p(\text{class} \mid \text{sky})$ whenever it occurs. Once sufficiently close to the minimum, the gradient will point "through the valley" towards amplifying the more relevant signal (Xing et al., 2018). However, this will also cause the sky feature to grow in magnitude—as well as its potential influence under selective perturbation, as described above. Both these factors contribute to progressive sharpening.

Here we emphasize the distinction between the loss on the _outliers_ and the full train loss. As images without sky (which comprise the majority of the dataset) are not nearly as sensitive to movement along this axis, the global loss landscape may not at first be significantly affected. Continued optimization will oscillate across the minimum with growing magnitude, but this growth may not be immediately apparent. Furthermore, _progress orthogonal to these oscillations need not be affected_—we find some evidence that these two processes occur somewhat independently, which we present in Section 4. Returning to the loss decomposition, we see that these oscillations will cause the losses to grow and alternate, with one group having high loss and then the other. Eventually the outliers' loss increases sufficiently and the overall loss spikes (Fig. 1), either flattening the valley or "catapulting" to a different basin (Wu et al., 2018; Lewkowycz et al., 2020; Thilak et al., 2022).

**Verifying our toy example's predictions.** Though this explanation lacks precise details, it does enable concrete predictions of network behavior during training. Fig. 4 tracks the predictions of a ResNet-18 on an image of sky—to eliminate possible confounders, we create a synthetic image as a

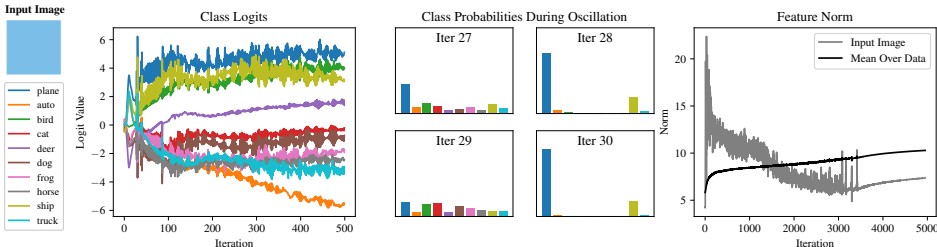

Figure 4: **Passing a sky-colored block through a ResNet during GD precisely tracks the predictions of our toy example. Left:** In the first phase, the network rapidly learns to use the sky feature to minimize loss. As signal is amplified, so too is the sky-colored input, and oscillation begins as depicted in Fig. 3. **Middle:** During oscillation, gradient steps alternate along the axis "sky = `plane`" (and a bit `ship`). **Right:** The initial phase and oscillation amplifies the sky input, as evidenced by the rapid growth in feature norm. The network then slowly learns to downweight this feature and rely on other signal (average feature norm provided for comparison).

single color block. Though the "`plane` vs. `other`" example seems almost *too* simple, we see exactly the described behavior—initial convergence to the minimum along with rapid growth in feature norm, followed by oscillation in class probabilities. Over time, the network learns to use other signal and downweights the sky feature, as evidenced by the slow decay in feature norm. We reproduce this figure for many other inputs and for a VGG-11-BN in Appendix C, with similar findings.

Our example also suggests that **oscillation serves as a valuable regularizer that reduces reliance on easily learned opposing signals which may not generalize.** When a signal is used to the benefit of one group and the detriment of another, the disadvantaged group's gradient will have larger magnitude, and the network will be encouraged to downweight the feature. In Appendix C.3 we reproduce Fig. 4 with a VGG-11-BN trained with a very small learning rate to closely approximate gradient flow. We see that gradient flow and GD are very similar until reaching the edge of stability. After this point, the feature norm under GD begins to slowly decay while oscillating; in contrast, in the absence of oscillation, the feature norms of opposing signals under gradient flow grow continuously. If opposing signals represent "simple" features which generalize worse, this could help to explain the poor generalization of gradient flow. A similar effect was observed by Jastrzębski et al. (2020), who noted that large initial learning rate leads to a better-conditioned loss landscape later.

### 3.3 THEORETICAL ANALYSIS OF OPPOSING SIGNALS IN A SIMPLE MODEL

To demonstrate this effect formally, we study misspecified linear regression on inputs $x \in \mathbb{R}^d$ with a two-layer linear network. Though this model is simplified, it enables preliminary insight into the factors we think are most important for these dynamics to occur. Since we are analyzing the dynamics from initialization until the stability threshold, it will be sufficient to study the trajectory of *gradient flow*—for reasonable step sizes $\eta$, a similar result then follows for gradient descent. Our analysis reproduces the initial phase of quickly reducing loss on the outliers, followed by the subsequent growth in sensitivity to the *way* the opposing signal is used—i.e., progressive sharpening. We also verify this pattern (and the subsequent oscillation, which we do not formally prove) in experiments on real and synthetic data in Appendix F.

**Model.** We model the observed features as a distribution over $x \in \mathbb{R}^{d_1}$, assuming only that its covariance $\Sigma$ exists—for clarity we treat $\Sigma = I$ in the main text. We further model a vector $x_o \in \mathbb{R}^{d_2}$ representing the opposing signal, with $d_2 > d_1$. We will suppose that on some small fraction of outliers $p \ll 1$, $x_o \sim \text{Unif}\left(\left\{\pm\sqrt{\frac{\alpha}{pd_2}}\mathbf{1}\right\}\right)$ ($\mathbf{1}$ is the all-ones vector) for some $\alpha$ which governs the feature magnitude, and we let $x_o = \mathbf{0}$ on the remainder of the dataset. We model the target as the function $y = \beta^\top x + \frac{1}{\sqrt{d_2}}\mathbf{1}^\top |x_o|$; this captures the idea that the signal $x_o$ correlates strongly with the target, but in opposing directions of equal strength. Finally, we parameterize the network with vectors $b \in \mathbb{R}^{d_1}, b_o \in \mathbb{R}^{d_2}$ and scalar $c$ in one single vector $\theta$, as $f_\theta(x) = c \cdot (b^\top x + b_o^\top x_o)$. Note the specific distribution of $x_o$ is unimportant for our analysis; only the mean and variance matter. Furthermore, in our simulations we observed the exact same behavior with cross-entropy loss. Our results suggest that depth and a small signal-to-noise ratio are the only elements needed for this behavior to arise.

**Setup.** A standard initialization would be to sample $[b, b_o]^\top \sim \mathcal{N}(0, \frac{1}{d_1+d_2}I)$, which would then imply highly concentrated distributions for the quantities of interest. As tracking the precise concentration terms would not meaningfully contribute to the analysis, we simplify by directly assuming that at initialization these quantities are equal to their expected order of magnitude: $\|b\|_2^2 = \mathbf{1}^\top b = \frac{d_1}{d_1+d_2}$, $\|b_o\|_2^2 = \mathbf{1}^\top b_o = \frac{d_2}{d_1+d_2}$, and $b^\top \beta = \frac{\|\beta\|}{\sqrt{d_1+d_2}}$. Likewise, we let $c = 1$, ensuring that both layers have the same norm. We perform standard linear regression by minimizing the population loss $L(\theta) := \frac{1}{2}\mathbb{E}[(f_\theta(x) - y)^2]$. We see that the minimizer of this objective has $b_o = \mathbf{0}$ and $cb = \beta$. However, an analysis of gradient flow will elucidate how depth and strong opposing signals lead to sharpening as this minimum is approached.

**Results.** In exploring progressive sharpening, Cohen et al. (2021) found that sometimes the model would have a brief *decrease* in sharpness, particularly for the square loss. In fact, this is consistent with our above explanation: for larger $\alpha$ and a sharper loss (e.g. the square loss), the network will initially prioritize minimizing loss on the outliers, thus heavily reducing sharpness. Our first result proves that this occurs in the presence of large magnitude opposing signals:

**Theorem 3.1** (Initial *decrease* in sharpness). *Let $k := \frac{d_2}{d_1}$, and assume $\|\beta\| > \max(\frac{d_1}{\sqrt{d_1+d_2}}, \frac{24}{5})$. At initialization, the sharpness $\|\nabla_\theta^2 L(\theta)\|_2$ lies in $(\alpha, 3\alpha)$. Further, if $\sqrt{\alpha} = \Omega(\|\beta\|k\ln k)$, then both $\|b_o\|_2^2$ and the overall sharpness will* decrease *as $\tilde{O}(e^{-\alpha t})$ from $t = 0$ until some time $t_1 \leq \frac{\ln \|\beta\|/2}{2\|\beta\|}$.*

Proofs can be found in Appendix G. After this decrease, signal amplification can proceed—but this also means that the sharpness with respect to *how the network uses the feature $x_o$* will grow, so a small perturbation to the parameters $b_o$ will induce a large increase in loss.

**Theorem 3.2** (Progressive sharpening). *If $\sqrt{\alpha} = \Omega(1 + \|\beta\|^2 k \ln k)$, then at starting at time $t_1$ the sharpness will increase linearly in $\|\beta\|$ until some time $t_2 \geq \frac{1}{2\|\beta\|_2^2}$, reaching at least $\frac{5}{8}\|\beta\|\alpha$. This lower bound on sharpness applies to each dimension of $b_o$.*

Oscillation will not occur during gradient flow—but for SGD with step size $\eta > \frac{16}{5\|\beta\|\alpha}$, $b_o$ will start to increase in magnitude while oscillating across the origin. If this growth continues, it will rapidly *reintroduce* the feature, causing the loss on the outliers to grow and alternate. Such reintroduction (an example of which occurs around iteration 3000 in Fig. 4) seems potentially helpful for exploration. In Fig. 40 in the Appendix we simulate our model and verify exactly this sequence of events.

## 3.4 ADDITIONAL FINDINGS

**Sharpness often occurs overwhelmingly in the first few layers.** In Appendix D we track what fraction of curvature[5] lies in each layer of various networks during training with GD. In a ResNet-18, sharpness occurrs almost exclusively in the first convolutional layer after the first few training steps; the same pattern appears more slowly while training a VGG-11. In a Vision Transformer curvature occurs overwhelmingly in the embedding layer and very slightly in the earlier MLP projection heads. GPT-2 (Radford et al., 2019) follows the same pattern, though with less extreme concentration in the embedding. Thus it does seem to be the case that earlier layers have the most significant sharpness— especially if they perform dimensionality reduction or have particular influence over how the signal is propagated to later layers. This seems the likely cause of large gradients in the early layers of vision models (Chen et al., 2021; Kumar et al., 2022), suggesting that this effect is equally influential during finetuning and pretraining and that further study can improve optimization.

**Batchnorm may smooth training, even if not the loss itself.** Cohen et al. (2021) noted that batchnorm (BN) (Ioffe & Szegedy, 2015) does not prevent networks from reaching the edge of stability and concluded, contrary to Santurkar et al. (2018), that BN does not smooth the loss landscape. We conjecture that the benefit of BN may be in downweighting the influence of opposing signals and mitigating this oscillation. In other words, BN may smooth the *optimization trajectory* of neural networks, rather than the loss itself (this is consistent with the distinction made by Cohen et al. (2021) between regularity and smoothness). In Section 4 we demonstrate that Adam *also* smooths

---

[5]The "fraction of curvature" is with respect to the top eigenvector of the loss Hessian. We partition this vector by network layer, so each sub-vector's squared norm represents that layer's contribution to the overall curvature.

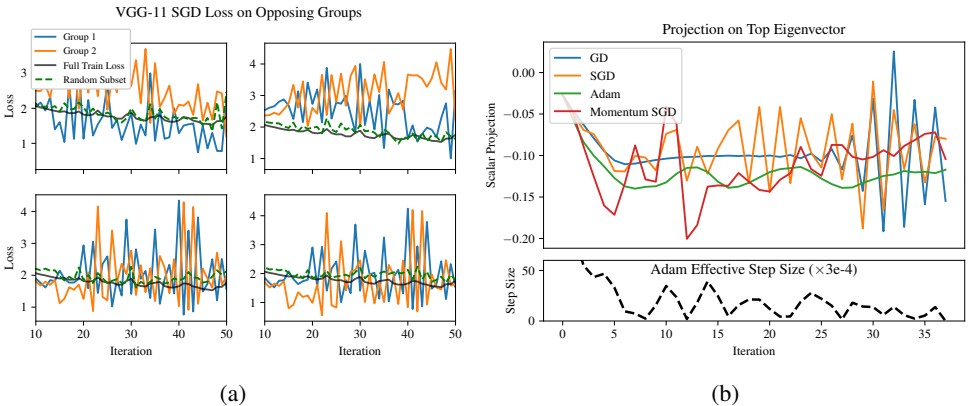

(a)                                                                        (b)

Figure 5: **Outliers with opposing signals have a significant influence even during SGD. Left:** We plot the losses of paired outlier groups on a VGG-11-BN trained on CIFAR-10, along with the full train loss for comparison. Modulo batch randomness, the outliers' loss follow the same oscillatory pattern with large magnitude. See appendix for the same without batchnorm. **Right (top):** We train a small MLP on a 5k subset of CIFAR-10 with various optimizers and project the iterates onto the top Hessian eigenvector. SGD closely tracks GD, bouncing across the valley; momentum somewhat mitigates the sharp jumps. Adam smoothly oscillates along one side. **Right (bottom):** Adam's effective step size drops sharply when moving too close or far from the valley floor.

the optimization trajectory and that minor changes to emulate this effect can aid stochastic optimization. We imagine that the effect of BN could also depend on the use of GD vs. SGD. Specifically, our findings hint at a possible benefit of BN which applies only to SGD: reducing the influence of imbalanced opposing signals in a random minibatch.

**For both GD and SGD, approximately half of training points go up in loss on each step.** Though only the outliers are wildly oscillating, many more images contain some small component of the features they exemplify. Fig. 37 in the Appendix shows that the fraction of points which increase in loss hovers around 50% for every step—to some extent, a small degree of oscillation appears to be happening to the entire dataset. Fig. 38 depicts the distribution of changes in loss of individual training points at each iteration: the density closely approximates a normal curve, with a mean consistently just below 0 and a standard deviation around 0.3-0.5.

## 4    THE INTERPLAY OF OPPOSING SIGNALS AND STOCHASTICITY

Full-batch GD is not used in practice when training NNs. It is therefore pertinent to ask what these findings imply about stochastic optimization. We begin by verifying that this pattern persists during SGD. Fig. 5(a) displays the losses for four opposing group pairs of a VGG-11-BN trained on CIFAR-10 with SGD batch size 128. We observe that the paired groups do exhibit clear opposing oscillatory patterns, but they do not alternate with every step, nor do they always move in opposite directions. This should not be surprising: we expect that not every batch will have a given signal in one direction or the other. For comparison, we include the *full* train loss in each figure—that is, including the points not in the training batch. We see that the loss on the outliers has substantially larger variance; to confirm that this is not just because the groups have many fewer samples, we also plot the loss on a random subset of training points of the same size. We reproduce this plot with a VGG-11 without BN in Fig. 39 in the Appendix.

Having verified that this behavior occurs in the stochastic setting, we conjecture that current best practices for neural network optimization owe much of their success to how they handle opposing signals. As a proof of concept, we will make this more precise with a preliminary investigation of the Adam optimizer (Kingma & Ba, 2014).

### 4.1    HOW ADAM HANDLES GRADIENTS WITH OPPOSING SIGNALS

To better understand their differences, Fig. 5(b) visualizes the parameter iterates of Adam and SGD with momentum on a ReLU MLP trained on a 5k subset of CIFAR-10, alongside those of GD and

SGD (all methods use the same initialization and sequence of training batches). The top figure is the projection of these parameters onto the top eigenvector of the loss Hessian of the network trained with GD, evaluated at the first step where the sharpness crosses $2/\eta$. We observe that SGD tracks a similar path to GD, though adding momentum mitigates the oscillation somewhat. In contrast, the network optimized with Adam markedly departs from this pattern, smoothly oscillating along one side. We identify three components of Adam which potentially contribute to this effect:

**Advantage 1: Smaller steps along directions with higher curvature.** Adam's normalization causes smaller steps along the top eigenvector, especially near the minimum. The lower plot in Fig. 5(b) shows that the effective step size in this direction—i.e., the absolute inner product of the parameter-wise step sizes and the top eigenvector—rapidly drops to zero as the iterates move orthogonal to the valley floor. Thus normalizing by curvature *parameter-wise* seems crucial; similarly, Pan & Li (2023) show that parameter-wise gradient clipping improves SGD substantially.

**Advantage 2: Managing heavy-tailed gradients and *avoiding* steepest descent.** Zhang et al. (2020) identified the "trust region" as important for Adam's success, pointing to heavy-tailed noise in the stochastic gradients. More recently, Kunstner et al. (2023) argued that Adam's superiority does not come from better handling noise, supported with large batch experiments. Our result reconciles these contradictory claims by showing that **the difficulty is not heavy-tailed *noise*, but strong, directed (and perhaps imbalanced) opposing signals.** Unlike traditional "gradient noise", larger batch sizes may not reduce this effect—that is, the full gradient is heavy-tailed across parameters. Furthermore, the largest steps emulate Sign SGD, which is notably *not* a descent method. Fig. 5(b) shows that Adam's steps are more parallel to the valley floor than those of steepest descent. **Thus it seems advantageous to *intentionally* avoid steepest descent,** which might lead to an ill-conditioned landscape. This point is also consistent with the observed generalization benefits of a large learning rate for SGD on NNs (Jastrzębski et al., 2020); in fact, opposing signals naturally fit the concept of "easy-to-fit" features as modeled by Li et al. (2019).

**Advantage 3: Dampening.** Traditional SGD with momentum $\beta < 1$ takes a step which weights the current gradient by $1/(1+\beta) > 1/2$. Though this makes intuitive sense, our results imply that heavily weighting the most recent gradient can be problematic. Instead, we expect an important addition is *dampening*, which multiplies the stochastic gradient at each step by some $(1 - \tau) < 1$. We observe that Adam's (unnormalized) gradient is equivalent to SGD with momentum and dampening both equal to $\beta_1$, plus a debiasing step. Recently proposed alternatives also include dampening in their momentum update but do not explicitly identify the distinction (Zhang et al., 2020; Pan & Li, 2023; Chen et al., 2023).

**Using these insights to aid stochastic optimization.** To test whether our findings translate to practical gains, we design a variant of SGD which incorporates these insights. First, we use dampening $\tau = 0.9$ in addition to momentum. Second, we choose a global threshold: if the gradient magnitude for a parameter is above this threshold, we take a fixed step size; otherwise, we take a gradient step as normal. The exact method appears in Appendix E. We emphasize that our goal here is not to propose a new optimization algorithm; we are exploring the potential value gained from knowledge of the existence and influence of opposing signals.

Results in Appendix E show that this approach matches Adam when training ResNet-56/110 on CIFAR-10 with learning rates for the unthresholded parameters across several orders of magnitude ranging from $10^{-4}$ to $10^3$. Notably, the fraction of parameters above the threshold is only around 10-25% per step, and the trajectory and behavior of the network is dominated by this small fraction; the remainder can be optimized much more robustly, but their effect on the network's behavior is obscured. We therefore see the influence of opposing signals as a possible explanation for the "hidden" progress in grokking (Barak et al., 2022; Nanda et al., 2023). We also compare this method to Adam for the early phase of training GPT-2 on the OpenWebText dataset (Gokaslan et al., 2019)—not only do they perform the same, their loss similarity suggests that their exact trajectory may be very similar (Appendix E.1). Here the fraction of parameters above the threshold hovers around 50% initially and then gradually decays. The fact that many more parameters in the transformer are above the threshold suggests that the attention mechanism is more sensitive to opposing signals and that further investigation of how to mitigate this instability may be fruitful.

## 5 CONCLUSION

The existence of outliers with such a significant yet non-obvious influence on neural network training raises as many questions as it answers. This work presents an initial investigation into their effect on various aspects of optimization, but there is still much more to understand. Though it is clear they have a large influence on training, less obvious is whether reducing their influence is *necessary* for improved optimization or simply coincides with it. At the same time, there is evidence that the behavior these outliers induce may serve as an important method of exploration and/or regularization. If so, another key question is whether these two effects can be decoupled—or if the incredible generalization ability of neural networks is somehow inherently tied to their instability.

## ACKNOWLEDGEMENTS

We thank Saurabh Garg for detailed feedback on an earlier version of this work. Thanks also to Christina Baek and Bingbin Liu for helpful comments and to Jeremy Cohen for pointers to related work. This research is supported in part by NSF awards IIS-2211907, CCF-2238523, an Amazon Research Award, and the CMU/PwC DT&I Center.

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

# A  RELATED WORK AND DISCUSSION

Many of the observations we make in this paper are not new, having been described in various prior works. Rather, this work identifies a possible *higher-order cause* which neatly ties these findings together. There are also many works which pursue a more theoretical understanding of each of these phenomena independently. Such analyses begin with a set of assumptions (on the data, in particular) and prove that the given behavior follows. In contrast, this work *begins* by identifying a condition—the presence of opposing signals—which we argue is likely a major cause of these behaviors. These two are not at odds: we believe in many cases our result serves as direct evidence for the validity of these modeling assumptions and that it may enable even more fine-grained analyses. This work provides an initial investigation which we hope will inspire future efforts towards a more complete understanding.

**Characterizing the NN loss landscape.**  Earlier studies of the loss landscape extensively explored the spectrum of the Hessian of the loss, with a common observation being a hierarchical structure with a small group of very large outlier eigenvalues (Sagun et al., 2016; 2017; Papyan, 2018; 2019; Fort & Ganguli, 2019; Ghorbani et al., 2019; Li et al., 2020; Papyan, 2020; Kopitkov & Indelman, 2020). Later efforts focused on concretely linking these observations to corresponding behavior, often with an emphasis on SGD's bias towards particular solutions (Wu et al., 2018; Jastrzębski et al., 2017; 2020) and what this may imply about its resulting generalization (Jastrzębski et al., 2019; Zhu et al., 2019; Wu et al., 2022). Our method for identifying these paired groups, along with Fig. 2, indicates that this outlier spectrum is precisely the directions with opposing signals in the gradient, and that this pattern may be key to better understanding the generalization ability of NNs trained with SGD.

**Progressive sharpening and the edge of stability.**  Shifting away from the overall structure, more recent focus has been specifically on top eigenvalue(s), where it was empirically observed that their magnitude (the loss "sharpness") grows when training with SGD (Jastrzębski et al., 2019; 2020) and GD (Kopitkov & Indelman, 2020; Cohen et al., 2021) (so-called "progressive sharpening"). This leads to rapid oscillation in weight space (Xing et al., 2018; Jastrzębski et al., 2019; Cohen et al., 2021; 2022). Cohen et al. (2021) also found that for GD this coincides with a consistent yet non-monotonic decrease in training loss over long timescales, which they named the "edge of stability"; moreover, they noted that this behavior runs contrary to our traditional understanding of NN convergence. Many works have since investigated the possible origins of this phenomenon (Zhu et al., 2023; Kreisler et al., 2023); several are deeply related to our findings. Ma et al. (2022) connect this behavior to the existence of multiple "scales" of losses; the outliers we identify corroborate this point. Damian et al. (2022) prove that GD implicitly regularizes the sharpness—we identify a conceptually distinct source of such regularization, as described in Section 3. Arora et al. (2022) show under some conditions that the GD trajectory follows a minimum-loss manifold towards lower curvature regions. This is consistent with our findings, and we believe this manifold to be precisely the path which evenly balances the opposing gradients. Wang et al. (2022) provide another thorough analysis of NN training dynamics at the edge of stability; their demonstrated phases closely align with our own. They further observe that this sharpening coincides with a growth in the norm of the last layer, which was also noted by MacDonald et al. (2023). Our proposed explanation for the effect of opposing signals offers some insight into this relationship.

Roughly, our results seem to imply that progressive sharpening occurs when the network learns to rely on (or *not* rely on) opposing signals in a very specific way, while simultaneously amplifying overall sensitivity. This growth in sensitivity means a small parameter change modifying how opposing signals are used can massively increase loss. This leads to intermittent instability orthogonal to the "valley floor", accompanied by gradual training loss decay and occasional spikes as described by the toy example in Fig. 3 and depicted on real data in Fig. 1.

**Generalization, grokking, slingshotting, and subnetworks.**  We see ties between opposing signals and many other phenomena in NN optimization, but a few specific concepts stand out to us as most clearly related. At least in images, the features which provide opposing signals match the traditional picture of "spurious correlations" surprisingly closely—we conjecture that the factors affecting whether a network maintains balance or diverges along a direction also determine whether it continues to use a "spurious" feature or is forced to find an alternative way to minimize loss.

Indeed, the exact phenomenon of a network "slingshotting" to a new region has been directly observed, along with a corresponding change in generalization (Wu et al., 2018; Lewkowycz et al., 2020; Jastrzębski et al., 2021; Thilak et al., 2022). "Grokking" (Power et al., 2022), whereby a network learns to generalize long after memorizing the training set, is another closely related area of study. Several works have shown that grokking is a "hidden" phenomenon, with gradual amplification of generalizing subnetworks (Barak et al., 2022; Nanda et al., 2023; Merrill et al., 2023); it has even been noted to co-occur with weight oscillation (Notsawo Jr et al., 2023). We observe that the opposing signal gradients dominate the overall gradient for most of training—and our experiments on SGD in Section 4 and Appendix E give some further evidence that NN optimization occurs on two different scales, obscured by the network's behavior on outliers. We note that this also relates to the Lottery Ticket Hypothesis (Frankle & Carbin, 2019): we think it is not unlikely that a pruned winning ticket is one which is not as influenced by opposing signals, allowing it to more easily traverse the loss landscape. In fact, said ticket may already be following this trajectory during standard optimization—but the behavior (and loss) of the remainder of the network on these opposing signals would obscure its progress.

**Simplicity bias and double descent.** Nakkiran et al. (2019) observed that NNs learn functions of increasing complexity throughout training. Our experiments—particularly the slow decay in the norm of the feature embedding of opposing signals—lead us to believe it would be more correct to say that they *unlearn* simple functions, which enables more complex subnetworks with smaller magnitude and better performance to take over. At first this seems at odds with the notion of *simplicity bias* (Valle-Perez et al., 2019; Shah et al., 2020), defined broadly as a tendency of networks to rely on simple functions of their inputs. However, it does seem to be the case that the network will use the simplest (e.g., largest norm) features that it can, so long as such features allow it to approach zero training loss; otherwise it may eventually diverge. This tendency also suggests a possible explanation for *double descent* (Belkin et al., 2019; Nakkiran et al., 2020): even after interpolation, the network pushes towards greater confidence and the weight layers continue to balance (Saxe et al., 2013; Du et al., 2018), increasing sharpness. This could lead to oscillation, pushing the network to learn new features which generalize better (Wu et al., 2018; Li et al., 2019; Rosenfeld et al., 2022; Thilak et al., 2022). This behavior would also be more pronounced for larger networks because they exhibit greater sharpening. Note that the true explanation is not quite so straightforward: generalization is sometimes improved via methods that *reduce* oscillation (like loss smoothing), implying that this behavior is not always advantageous. A better understanding of these nuances is an important subject for future study.

**Sharpness-Aware Minimization** Another connection we think merits further inquiry is Sharpness-Aware Minimization (SAM) (Foret et al., 2021), which is known to improve generalization of neural networks for reasons still not fully understood (Wen et al., 2023). In particular, the better-performing variant is 1-SAM, which takes positive gradient steps on each training point in the batch individually. It it evident that several of these updates will point along directions of steepest descent/ascent orthogonal to the valley floor (and, if not normalized, the updates may be *very* large). Thus it may be that 1-SAM is in some sense "simulating" oscillation and divergence out of this valley in both directions, enabling exploration in a manner that would not normally be possible until the sharpness grows large enough—these intermediate steps would also encourage the network to downweight these features sooner and faster. In contrast, standard SAM would only take this step in one of the two directions, or perhaps not at all if the opposing signals are equally balanced. Furthermore, unlike 1-SAM the intermediate step would blend together all opposing signals in the minibatch. These possibilities seem a promising direction for further exploration.

## B  EXAMPLES OF OPPOSING SIGNALS IN TEXT

---

**Punctuation Ordering**

```
the EU is \the best war-avoidance mechanism ever invented["].
because it was one of the few that still \dry-farmed["].
He describes the taste as \almost minty["].
I did receive several offers to \help out a bit["].
Nor is it OK to say \the real solution is in a technological
breakthrough["].
and that's what they mean by \when complete["].
and that the next big investment bubble to burst is the \carbon
bubble["].
he had been \driven by ideological and political motives["].
Prime Minister Najib Razak's personal bank account was a \genuine
donation["].
exceptional intellect, unparalleled integrity, and record of
independence["].
was the \most consequential decision I've ever been involved
with["].
which some lawmakers have called the \filibuster of all
filibusters["].
Democrats vowed to filibuster what some openly called a \stolen
seat["].
```
- - - - - - - - - - - - - - - - - - - - - - - - - - - - - - - - - - - - - - - - -
```
His leather belt was usually the delivery method of choice.["]
\It's laborious and boring.  He loved excitement and attention.["]
'That little son-of-a-gun is playing favorites,' and turned it
against him.["]
\For medicinal purposes, for medical purposes, absolutely, it's
fine.["]
\That's a choice that growers make.  It's on their side of the
issue.["]
to be a good steward of your land.  You have to make big decisions
in a hurry.["]
\But of course modern farming looks for maximum yield no matter
what you have to put in.  And in the case of California, that input
is water.["]
And we have been drawing down on centuries of accumulation.  Pretty
soon those systems are not going to be able to provide for us.["]
not a luxury crop like wine.  I'm really excited ab out the
potential.["]
\We knew and still believe that it was the right thing to do.["]
spending lots of time in the wind tunnel, because it shows when we
test them.["]
Compliance was low on the list, but I think it's a pretty
comfortable bike.["]
the opportunity to do that.'  I just needed to take it and run with
it.["]
```

Figure 6: **Examples of opposing signals in text.** Found by training GPT-2 on a subset of OpenWeb-Text. Sequences are on separate lines, the token in brackets is the target and all prior tokens are (the end of the) context. As both standards are used, it is not always clear whether punctuation will come before or after the end of a quotation (we include the period after the quote for clarity—the model does not condition on it). Note that the double quotation is encoded as the *pair* of tokens [447, 251], and the loss oscillation is occurring for sequences that end with this pair, either before (top) or after (bottom) the occurrence of the period token (13).

---

**New Line or 'the' After Colon**

```
In order to prepare your data, there are three things to do:[\n]
in the FP lib of your choice, namely Scalaz or Cats.  It looks like
this:[\n]
Salcedo said of the work:[\n]
Enter your email address:[\n]
According to the CBO update:[\n]
Here's a list of 5 reasons as to why self diagnosis is valid:[\n]
successive Lambda invocations.  It looks more or less like
this:[\n]
data, there are three things to do:[\n]
4.2 percent in early 2018.\n\nAccording to the CBO update:[\n]
other than me being myself."\n\nWATCH:[\n]
is to make the entire construction plural.\n\nTwo recent
examples:[\n]
We offer the following talking points to anyone who is attending
the meeting:[\n]
is on the chopping block - and at the worst possible moment:[\n]
```
- - - - - - - - - - - - - - - - - - - - - - - - - - - - - - - - - - - - - - - -
```
as will our MPs in Westminster.  But to me it is obvious:  [the]
The wheelset is the same as that on the model above:  [the]
not get so engrained or in a rut with what I had been doing.  Not
to worry:  [the]
robs this incredible title of precisely what makes it so wonderful:
[the]
you no doubt noticed something was missing:  [the]
Neil Gorsuch's 'sexist' comments on maternity leave:  [the]
```

Figure 7: **Examples of opposing signals in text.** Found by training GPT-2 on a subset of Open-WebText. Sequences are on separate lines, the token in brackets is the target and all prior tokens are (the end of the) context. Sometimes a colon occurs mid-sentence—and is often followed by "the"—other times it announces the start of a new line. The model must *unlearn* ": $\mapsto$ [\n]" versus ": $\mapsto$ [the]" and instead use other contextual information.

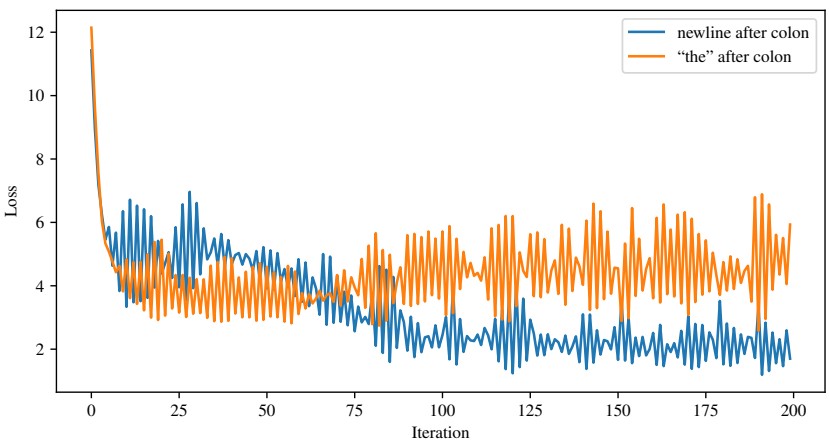

Figure 8: Loss of GPT-2 on the above opposing signals.

# C    REPRODUCING FIG. 4 IN OTHER SETTINGS

Though colors are straightforward, for some opposing signals such as grass texture it is not clear how to produce a synthetic image which properly captures what precisely the model is latching on to. Instead, we identify a real image which has as much grass and as little else as possible, with the understanding that the additional signal in the image could affect the results. We depict the grass image alongside the plots it produced.

## C.1    RESNET-18 TRAINED WITH GD ON OTHER INPUTS

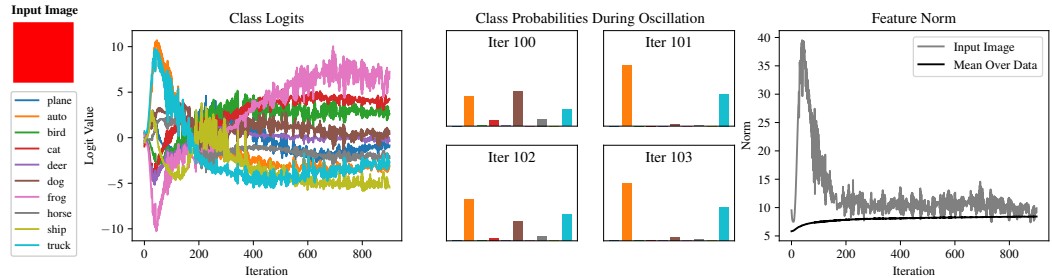

Figure 9: ResNet-18 on a red color block.

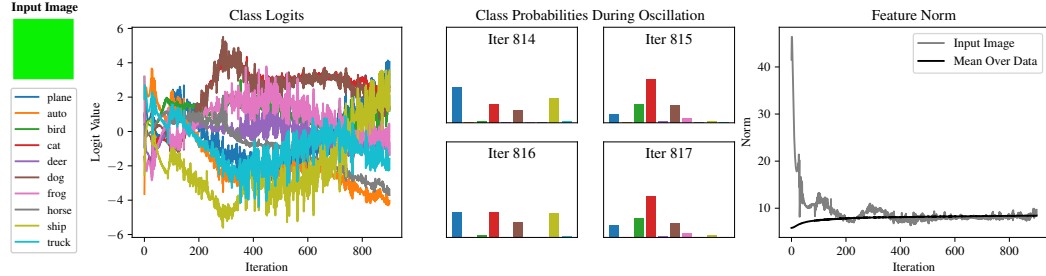

Figure 10: ResNet-18 on a green color block. As this color seems unnatural, we've included two examples of relevant images in the dataset.

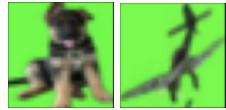

Figure 11: Examples of images with the above green color.

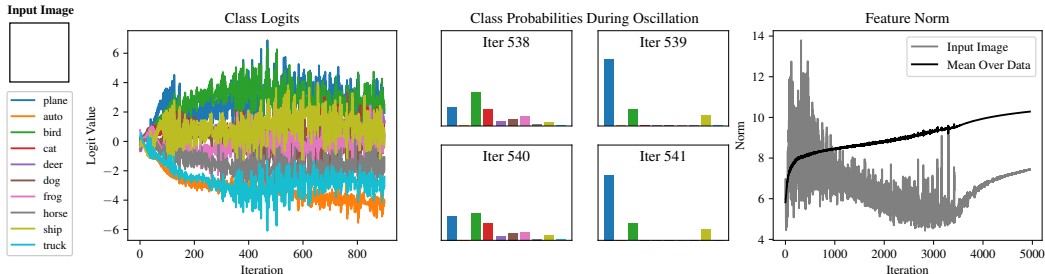

Figure 12: ResNet-18 on a white color block.

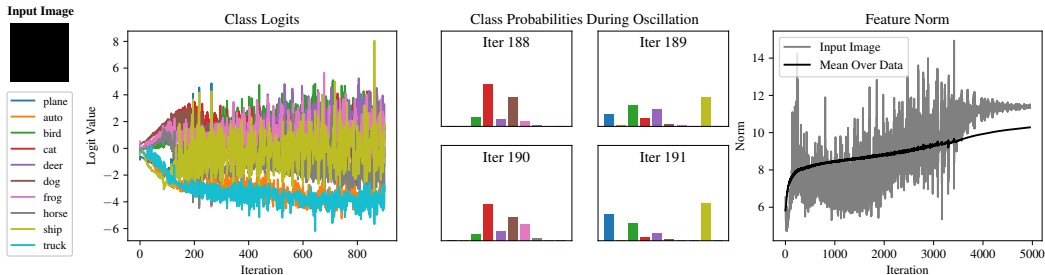

Figure 13: ResNet-18 on a black color block.

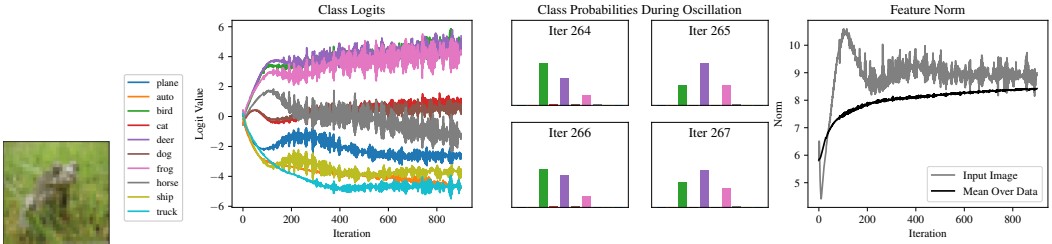

Figure 14: ResNet-18 on an image with mostly grass texture.

## C.2 VGG-11-BN TRAINED WITH GD

For VGG-11, we found that the feature norm of the embedded images did not decay nearly as much over the course of training. We expect this has to do with the lack of a residual component. However, for the most part these features do still follow the pattern of a rapid increase, followed by a marked decline.

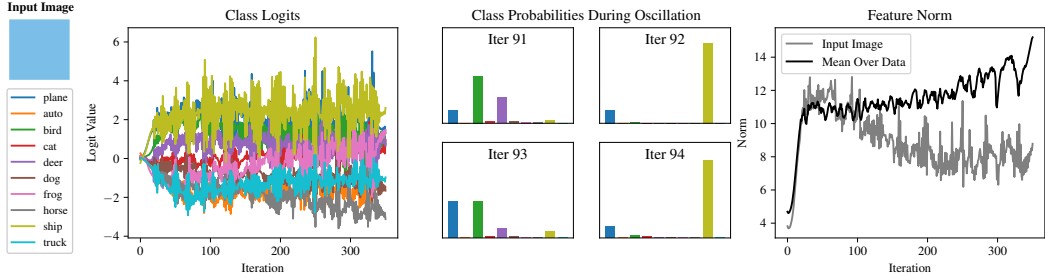

Figure 15: VGG-11-BN on a sky color block.

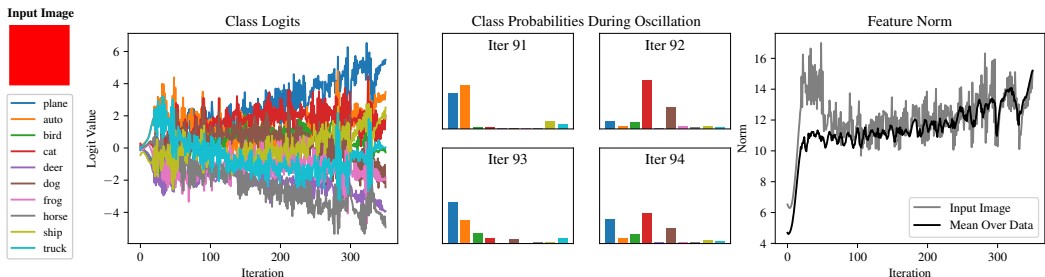

Figure 16: VGG-11-BN on a red color block.

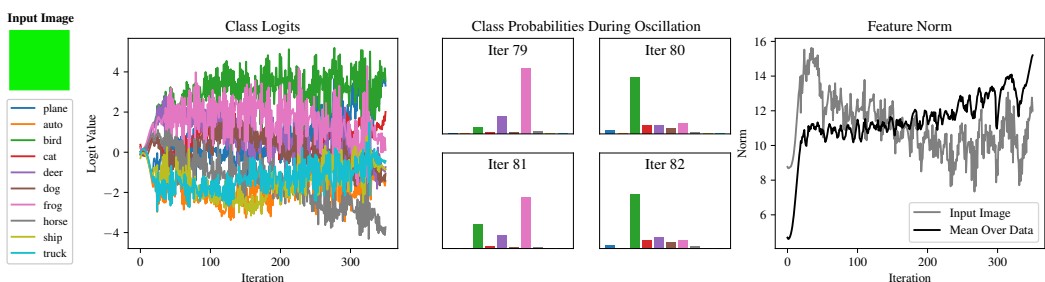

Figure 17: VGG-11-BN on a green color block. See above for two examples of relevant images in the dataset.

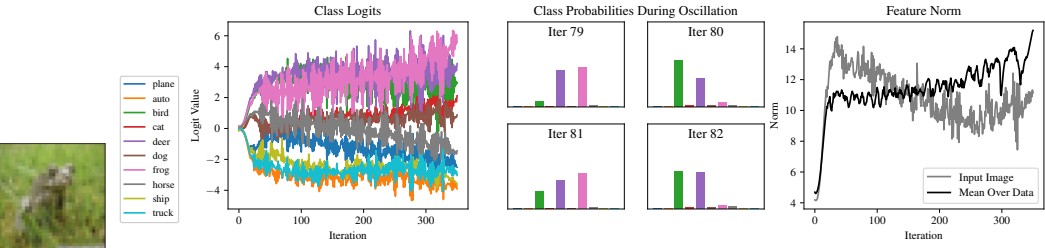

Figure 18: VGG-11-BN on an image with mostly grass texture.

### C.3 VGG-11-BN with Small Learning Rate to Approximate Gradient Flow

Here we see that oscillation is a valuable regularizer, preventing the network from continuously upweighting opposing signals. As described in the main body, stepping too far in one direction causes an imbalanced gradient between the two opposing signals. Since the group which now has a larger loss is also the one which suffers from the use of the feature, the network is encouraged to downweight its influence. If we use a very small learning rate to approximate gradient flow, this regularization does not occur and the feature norms grow continuously. This leads to over-reliance on these features, suggesting that failing to downweight opposing signals is a likely cause of the poor generalization of networks trained with gradient flow.

The following plots depict a VGG-11-BN trained with learning rate .0005 to closely approximate gradient flow. We compare this to the feature norms of the same network trained with gradient descent with learning rate 0.1, which closely matches gradient flow until it becomes unstable..

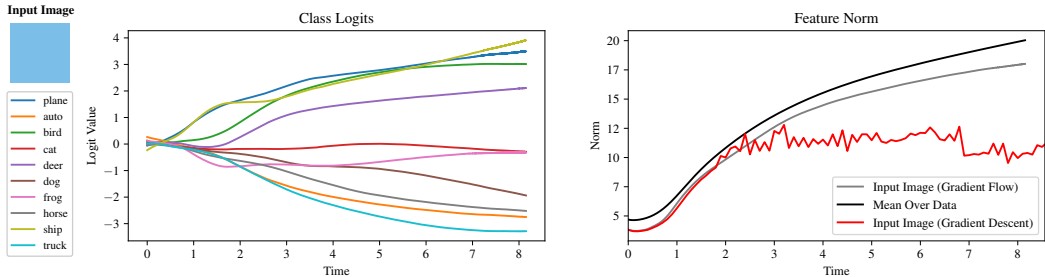

Figure 19: VGG-11-BN on a sky color block with learning rate 0.005 (approximating gradient flow) compared to 0.1.

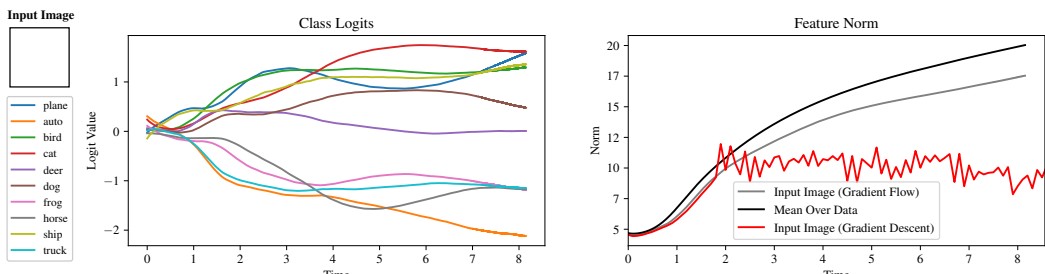

Figure 20: VGG-11-BN on a white color block with learning rate 0.005 (approximating gradient flow) compared to 0.1.

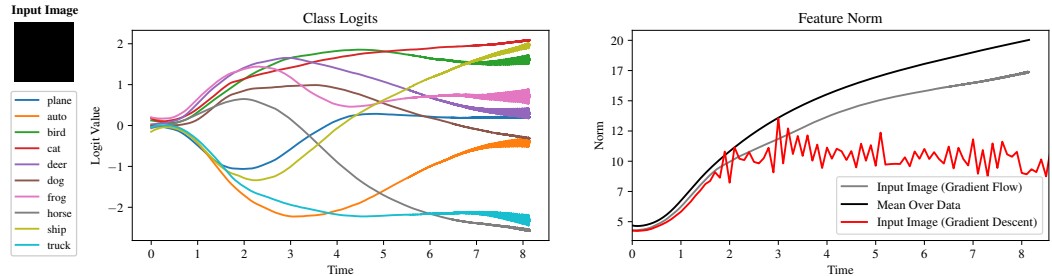

Figure 21: VGG-11-BN on a black color block with learning rate 0.005 (approximating gradient flow) compared to 0.1.

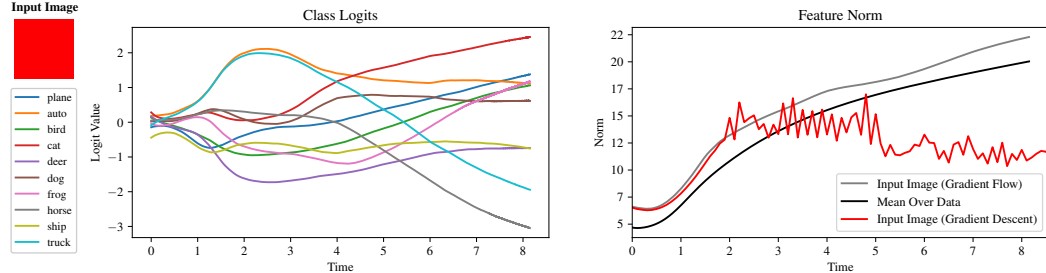

Figure 22: VGG-11-BN on a red color block with learning rate 0.005 (approximating gradient flow) compared to 0.1.

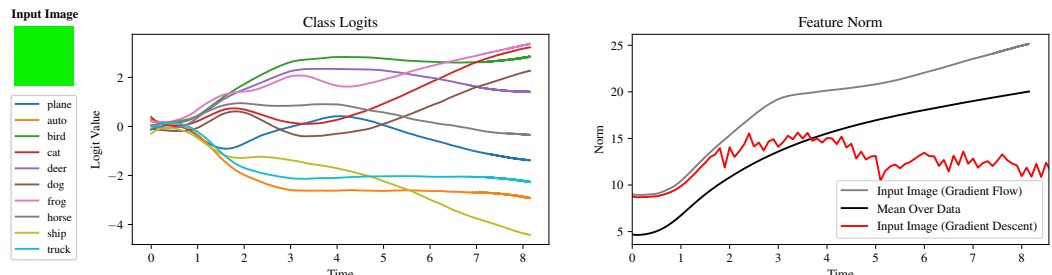

Figure 23: VGG-11-BN on a green color block with learning rate 0.0005 (approximating gradient flow) compared to 0.1.

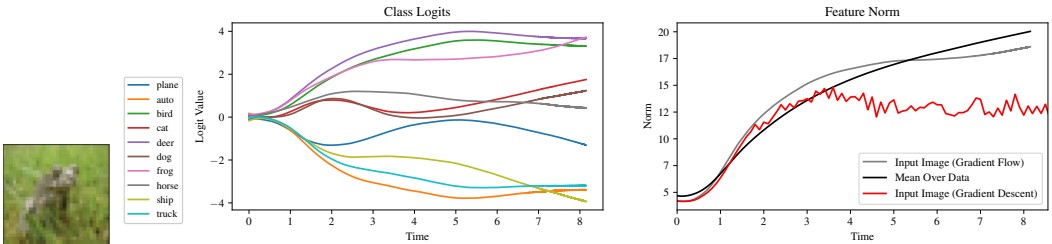

Figure 24: VGG-11-BN on an image with mostly grass texture with learning rate 0.0005 (approximating gradient flow) compared to 0.1.

## C.4 RESNET-18 TRAINED WITH FULL-BATCH ADAM

Finally, we plot the same figures for a ResNet-18 trained with full-batch Adam. We see that Adam consistently and quickly reduces the norm of these features, especially for more complex features such as texture, and that it also quickly reaches a point where oscillation ends. Note when comparing to plots above that the maximum iteration on the x-axis differs.

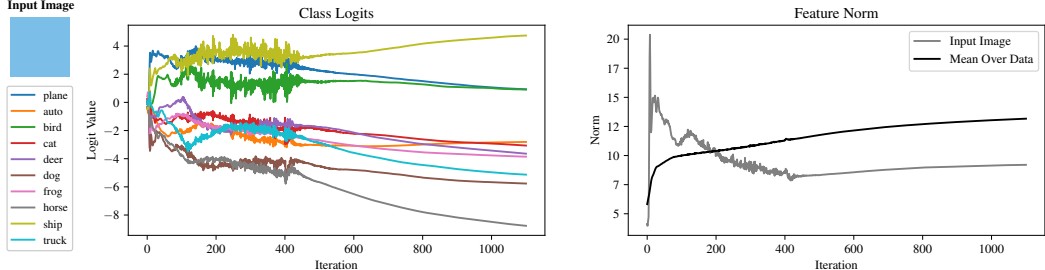

Figure 25: ResNet-18 on a sky color block trained with Adam.

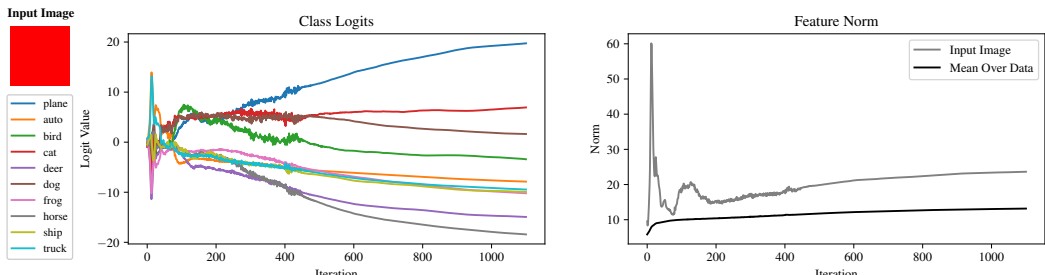

Figure 26: ResNet-18 on a red color block trained with Adam.

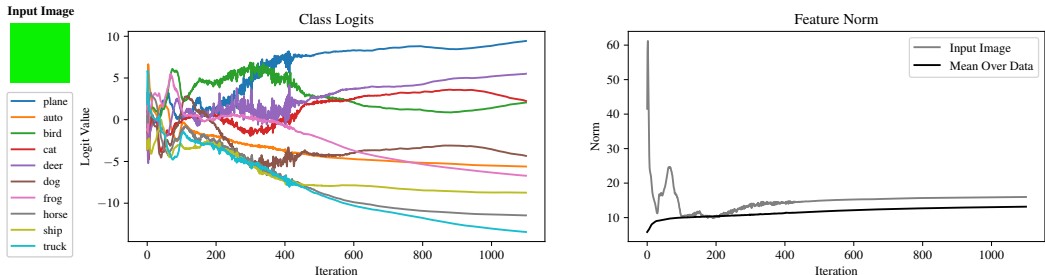

Figure 27: ResNet-18 on a green color block trained with Adam.

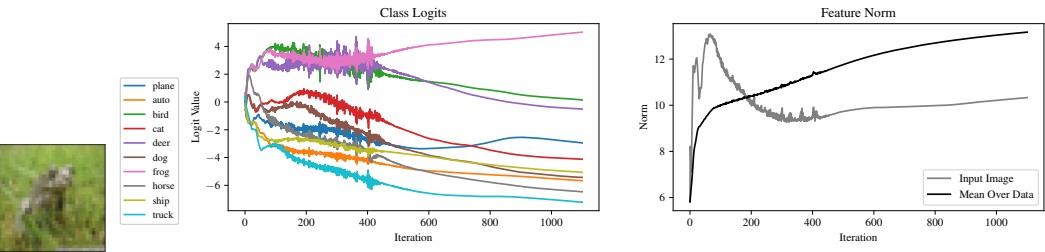

Figure 28: ResNet-18 on an image with mostly grass texture trained with Adam.

# D    TRACKING THE AMOUNT OF CURVATURE IN EACH PARAMETER LAYER

Here we plot the "fraction of curvature" of different architectures at various training steps. Recall the fraction of curvature is defined with respect to the top eigenvector of the loss Hessian. We partition this vector by network layer and evaluate each sub-vector's squared norm. This represents that layer's contribution to the overall curvature. To keep the plots readable, we omit layers whose fraction is never greater than 0.01 at any training step (including the intermediate ones not plotted), though we always include the last linear layer. The total number of layers is 45, 38, 106, and 39 for the ResNet, VGG-11, ViT, and NanoGPT respectively.

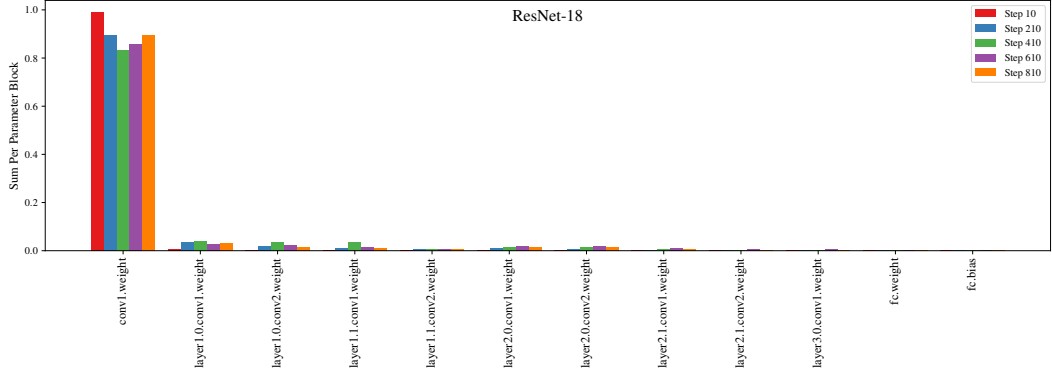

Figure 29: Sum of squared entries of the top eigenvector of the loss Hessian which lie in each parameter layer of a ResNet-18 throughout training.

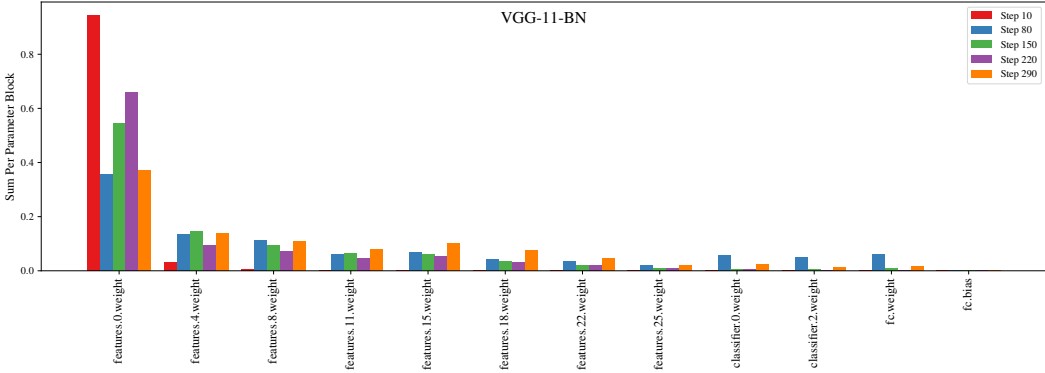

Figure 30: Sum of squared entries of the top eigenvector of the loss Hessian which lie in each parameter layer of a VGG-11-BN throughout training.

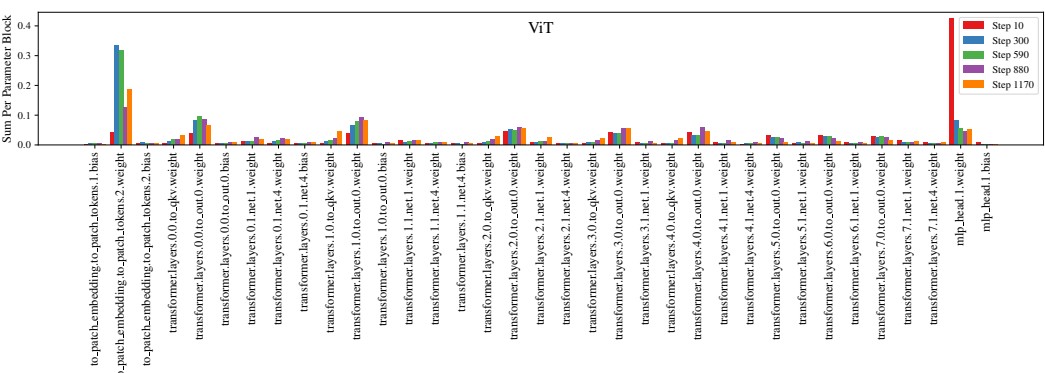

Figure 31: Sum of squared entries of the top eigenvector of the loss Hessian which lie in each parameter layer of a ViT throughout training.

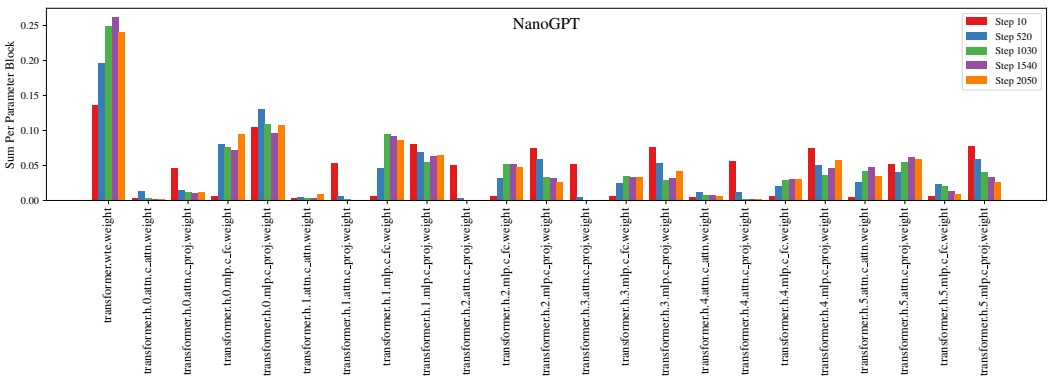

Figure 32: NanoGPT (6 layers, 6 head per layer, embedding dimension 384) trained on the default shakespeare character dataset in the NanoGPT repository. Due to difficulty calculating the true top eigenvector, we approximate it with the exponential moving average of the squared gradient.

# E    COMPARING OUR VARIANT OF SGD TO ADAM

As described in the main text, we find that simply including dampening and taking a fixed step size on gradients above a certain threshold results in performance matching that of Adam for the experiments we tried. We found that setting this threshold equal to the $q = .1$ quantile of the very first gradient worked quite well—this was about 1e-4 for the ResNet-110 and 1e-6 for GPT-2.

For the ResNet we also ablate the use of dampening: we find that masking appears to be much more important for early in training, while dampening is helpful for maintaining performance later. It is not immediately what may be the cause of this, nor if it will necessarily transfer to attention models.

Simply to have something to label it with, we name the method SplitSGD, because it performs SGD and SignSGD on different partitions of the data. We emphasize that we do not intend to introduce this as a new optimization algorithm or assert any kind of superior performance—our goal is only to demonstrate the insight gained from knowledge of opposing signals' influence on NN optimization.

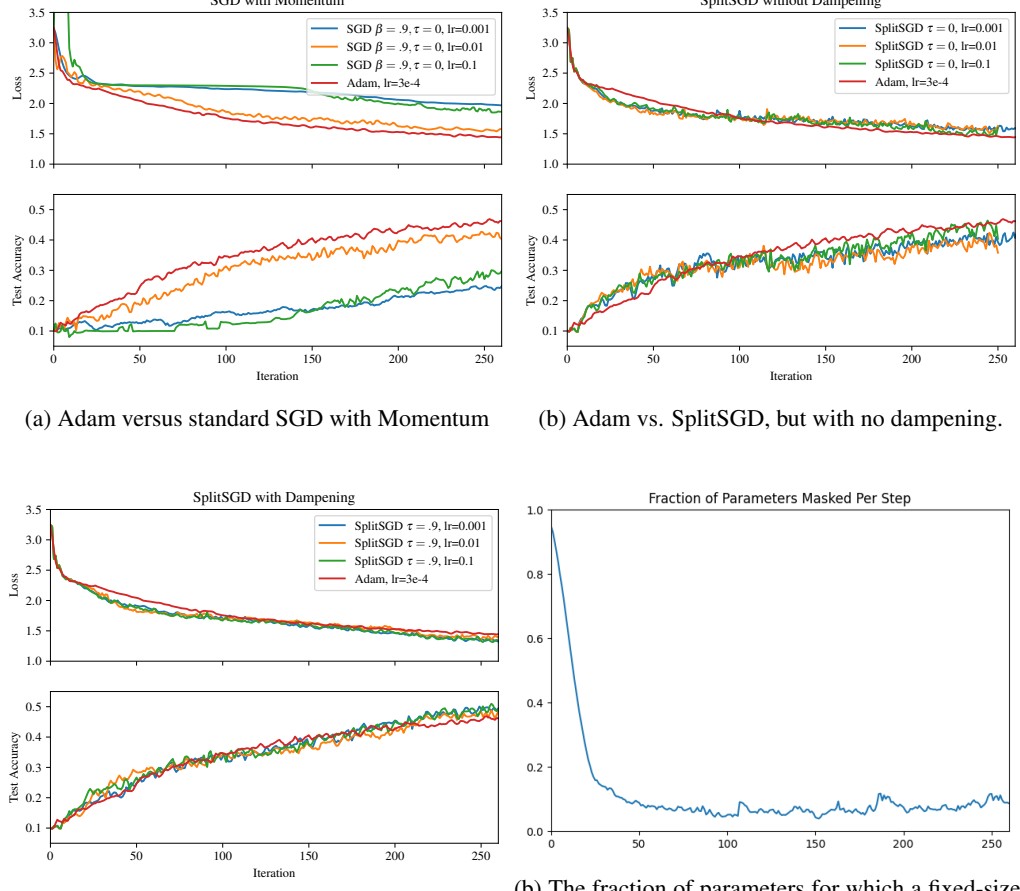

(a) Adam versus standard SGD with Momentum

(b) Adam vs. SplitSGD, but with no dampening.

(a) Adam vs. SplitSGD with dampening

(b) The fraction of parameters for which a fixed-size signed step was taken for each gradient step.

## E.1    SPLITSGD ON GPT-2

For the transformer, we use the public nanoGPT repository which trains GPT-2 on the OpenWebText dataset. As a full training run would be too expensive, we compare only for the early stage of optimization. Not only do the two methods track each other closely, it appears that they experience *exactly* the same oscillations in their loss. Though we do not track the parameters themselves, we believe it would be meaningful to investigate if these two methods follow very similar optimization trajectories as well.

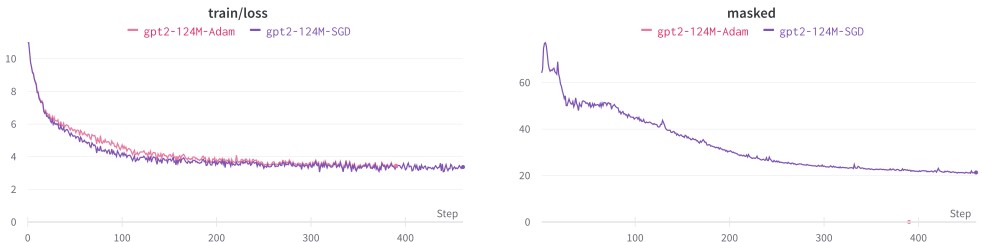

Figure 35: Adam versus SplitSGD on the initial stage of training GPT-2 on the OpenWebText dataset, and the fraction of parameters with a fixed-size signed step. All hyperparameters are the defaults from the nanoGPT repository. Observe that not only is their performance similar, they appear to have *exactly* the same loss oscillations.

# F  ADDITIONAL FIGURES

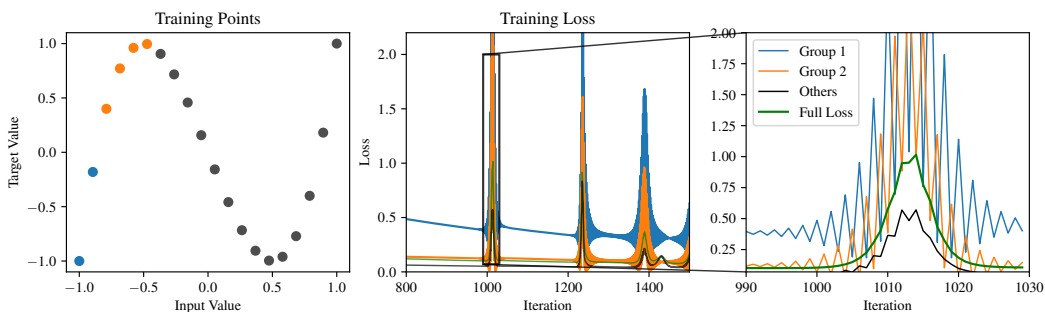

Figure 36: Opposing signals when fitting a Chebyshev polynomial with a small MLP. Though the data lacks traditional "outliers", it is apparent that the network has some features which are most influential only on the more negative inputs (or whose effect is otherwise cancelled out by other features). Since the correct use of this feature is opposite for these two groups, they provide opposing signals.

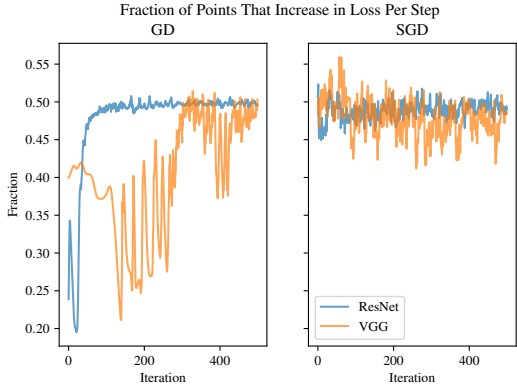

Figure 37: The fraction of overall training points which increase in loss on any given step. For both SGD and GD, it hovers around 0.5 (VGG without batchnorm takes a long time to reach the edge of stability). Though the outliers have much higher amplitude in their loss change, many more images contain *some* small component of the features they exemplify (or are otherwise slightly affected by the weight oscillations), and so these points also oscillate in loss at a smaller scale.

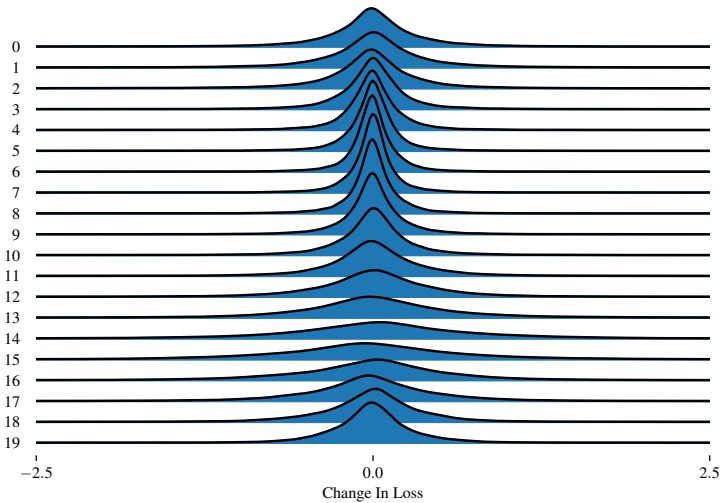

Figure 38: Distribution of changes in loss of individual training points at each iteration (starting at iteration 200 on a ResNet-18). The density is a nice bell curve, with consistent mean near 0 and standard deviation in $[0.3, 0.5]$, sometimes a bit more or less. The fraction of samples at a loss change greater in magnitude than $\mu \pm c\sigma$ is consistently very near to $[0.055, 0.015, 0.005]$ for $c \in [2, 3, 4]$.

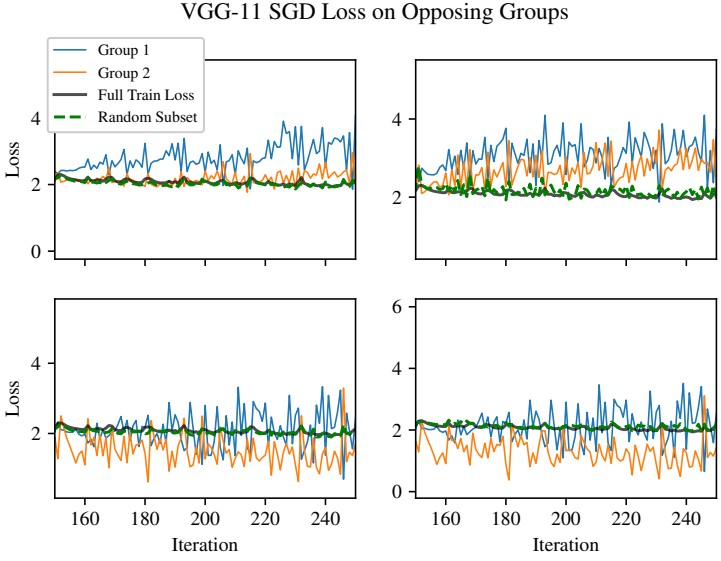

Figure 39: We reproduce Fig. 5(a) without batch normalization.

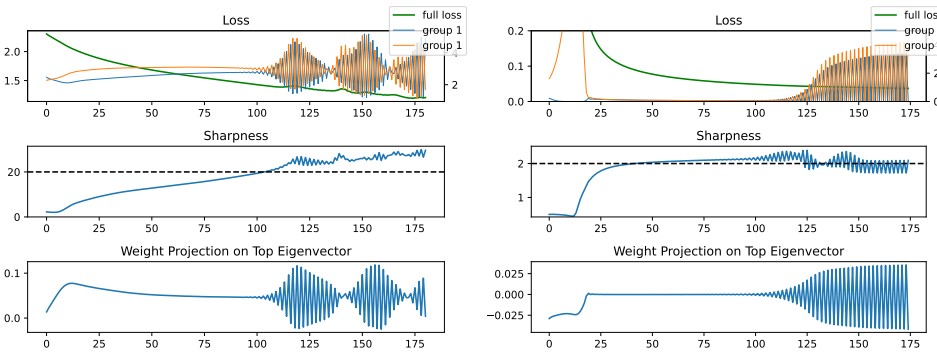

(a) A 3-layer ReLU MLP trained on a 5k-subset of CIFAR-10.

(b) Our model: a 2-layer linear network trained on mostly Gaussian data with opposing signals.

Figure 40: We compare a small ReLU MLP on a subset of CIFAR-10 to our simple model of linear regression with a two-layer network.

## G    PROOFS OF THEORETICAL RESULTS

Before we begin the analysis, we must identify the quantities of interest during gradient flow and the system of equations that determines how they evolve.

We start by writing out the loss:

$$2L(\theta) = \mathbb{E}[(c(b^\top x + b_o^\top x_o) - (\beta^\top x + d_2^{-1/2}\mathbf{1}^\top |x_o|))^2] \tag{1}$$

$$= \mathbb{E}[((cb - \beta)^\top x)^2] + \mathbb{E}[((cb_o - d_2^{-1/2}\operatorname{sign}(x_o)\mathbf{1})^\top x_o)^2] \tag{2}$$

$$= \|cb - \beta\|^2 + \frac{p}{2}\left(\left(\sqrt{\frac{\alpha}{p}}(cb_o - 1)\right)^2 + \left(\sqrt{\frac{\alpha}{p}}(cb_o + 1)\right)^2\right) \tag{3}$$

$$= \|cb - \beta\|^2 + \alpha(c^2\|b_o\|^2 + 1). \tag{4}$$

This provides the gradients

$$\nabla_b L = c(cb - \beta), \tag{5}$$

$$\nabla_{b_o} L = \alpha c^2 b_o, \tag{6}$$

$$\nabla_c L = b^\top(cb - \beta) + \alpha\|b_o\|^2 c. \tag{7}$$

We will also make use of the Hessian to identify its top eigenvalue; it is given by

$$\nabla_\theta^2 L(\theta) = \begin{bmatrix} c^2 I_{d_1} & \mathbf{0}_{d_1 \times d_2} & 2cb \\ \mathbf{0}_{d_2 \times d_1} & \alpha c^2 I_{d_2} & 2c\alpha b_o \\ 2cb^\top & 2c\alpha b_o^\top & \|b\|^2 + \alpha\|b_o\|^2 \end{bmatrix}. \tag{8}$$

The maximum eigenvalue $\lambda_{\max}$ at initialization is upper bounded by the maximum row sum of this matrix, and thus $\lambda_{\max} \leq 3\frac{d_1 + \alpha d_2}{d_1 + d_2} < 3\alpha$. Clearly, we also have $\lambda_{\max} \geq \alpha$.

We observe that tracking the precise vectors $b, b_o$ are not necessary to uncover the dynamics when optimizing this loss. First, let us write $b := \epsilon\frac{\beta}{\|\beta\|} + \delta v$, where $v$ is the direction of the rejection of $b$ from $\beta$ (i.e., $\beta^\top v = 0$) and $\delta$ is its norm. Then we have the gradients

$$\nabla_\epsilon L = (\nabla_\epsilon b)^\top(\nabla_b L) \tag{9}$$

$$= \frac{\beta}{\|\beta\|}^\top\left(c^2\left(\epsilon\frac{\beta}{\|\beta\|} + \delta v\right) - c\beta\right) \tag{10}$$

$$= c^2\epsilon - c\|\beta\|, \tag{11}$$

$$\nabla_\delta L = (\nabla_\delta b)^\top(\nabla_b L) \tag{12}$$

$$= v^\top\left(c^2\left(\epsilon\frac{\beta}{\|\beta\|} + \delta v\right) - c\beta\right) \tag{13}$$

$$= c^2\delta, \tag{14}$$

$$\nabla_c L = \left(\epsilon\frac{\beta}{\|\beta\|} + \delta v\right)^\top\left(c\left(\epsilon\frac{\beta}{\|\beta\|} + \delta v\right) - \beta\right) + \alpha\|b_o\|^2 c \tag{15}$$

$$= c(\epsilon^2 + \delta^2 + \alpha\|b_o\|^2) - \epsilon\|\beta\|. \tag{16}$$

Finally, define the scalar quantity $o := \|b_o\|^2$, noting that $\nabla_o L = 2b_o^\top \nabla_{b_o} L = 2\alpha c^2 o$. Minimizing this loss via gradient flow is therefore characterized by the following ODE on four scalars:

$$\frac{d\epsilon}{dt} = -c^2\epsilon + c\|\beta\|, \tag{17}$$

$$\frac{d\delta}{dt} = -c^2\delta, \tag{18}$$

$$\frac{do}{dt} = -2\alpha c^2 o, \tag{19}$$

$$\frac{dc}{dt} = -c(\epsilon^2 + \delta^2 + \alpha o) + \epsilon\|\beta\|. \tag{20}$$

$$\tag{21}$$

Furthermore, we have the boundary conditions

$$\epsilon(0) = \sqrt{\frac{1}{d_1 + d_2}}, \tag{22}$$

$$\delta(0) = \sqrt{\frac{d_1 - 1}{d_1 + d_2}}, \tag{23}$$

$$o(0) = \frac{d_2}{d_1 + d_2}, \tag{24}$$

$$c(0) = 1. \tag{25}$$

Given these initializations and dynamics, we make a few observations: (i) all four scalars are initialized at a value greater than 0, and remain greater than 0 at all time steps; (ii) $\delta$ and $o$ will decrease towards 0 monotonically, and $\epsilon$ will increase monotonically until $c\epsilon = \|\beta\|$; (iii) $c$ will be decreasing at initialization. Lastly, we define the quantity $r := (\epsilon(0)^2 + \delta(0)^2 + \alpha o(0)) = \frac{d_1 + \alpha d_2}{d_1 + d_2}$ and $k := \frac{d_2}{d_1}$.

Before we can prove the main results, we present a lemma which serves as a key tool for deriving continuously valid bounds on the scalars we analyze:

**Lemma G.1.** *Consider a vector valued ODE with scalar indices $v_1, v_2, \ldots,$ where each index is described over the time interval $[t_{\min}, t_{\max}]$ by the continuous dynamics $\frac{dv_i(t)}{dt} = a_i(v_{-i}(t)) \cdot v_i(t) + b_i(v_{-i}(t))$ with $a_i \leq 0, b_i \geq 0$ for all $i, t$ ($v_{-i}$ denotes the vector $v$ without index $i$). That is, each scalar's gradient is an affine function of that scalar with a negative coefficient. Suppose we define continuous functions $\hat{a}_i, \hat{b}_i : \mathbb{R} \to \mathbb{R}$ such that $\forall i, t, \hat{a}_i(t) \leq a_i(v_{-i}(t))$ and $\hat{b}_i(t) \leq b_i(v_{-i}(t))$. Let $\hat{v}$ be the vector described by these alternate dynamics, with the boundary condition $\hat{v}_i(t_{\min}) = v_i(t_{\min})$ and $v_i(t_{\min}) \geq 0$ for all $i$ (if a solution exists). Then for $t \in [t_{\min}, t_{\max}]$ it holds that*

$$\hat{v}(t) \leq v(t), \tag{26}$$

*elementwise. If $\hat{a}_i, \hat{b}_i$ upper bound $a_i, b_i$, the inequality is reversed.*

*Proof.* Define the vector $w(t) := \hat{v}(t) - v(t)$. This vector has the dynamics

$$\frac{dw_i}{dt} = \frac{d\hat{v}_i}{dt} - \frac{dv_i}{dt} \tag{27}$$

$$= \hat{a}_i(t) \cdot \hat{v}_i(t) + \hat{b}_i(t) - a_i(v_{-i}(t)) \cdot v_i(t) - b_i(v_{-i}(t)) \tag{28}$$

$$\leq \hat{a}_i(t) \cdot \hat{v}_i(t) - a_i(v_{-i}(t)) \cdot v_i(t). \tag{29}$$

The result will follow by showing that $w(t) \leq \mathbf{0}$ for all $t \in [t_{\min}, t_{\max}]$ (this clearly holds at $t_{\min}$). Assume for the sake of contradiction there exists a time $t' \in (t_{\min}, t_{\max}]$ and index $i$ such that $w_i(t') > 0$ (let $i$ be the first such index for which this occurs, breaking ties arbitrarily). By continuity, we can define $t_0 := \max \{t \in [t_{\min}, t'] : w_i(t) \leq 0\}$. By definition of $t_0$ it holds that $w_i(t_0) = 0$ and $\forall \epsilon > 0, w_i(t_0 + \epsilon) - w_i(t_0) = w_i(t_0 + \epsilon) > 0$, and thus $\frac{dw_i(t_0)}{dt} > 0$. But by the definition of $w$ we also have

$$\hat{v}_i(t_0) = v_i(t_0) + w_i(t_0) \tag{30}$$

$$= v_i(t_0), \tag{31}$$

and therefore

$$\frac{dw_i(t_0)}{dt} \leq \hat{a}_i(t_0) \cdot \hat{v}_i(t_0) - a_i(v_{-i}(t_0)) \cdot v_i(t_0) \tag{32}$$

$$= \left(\hat{a}_i(t_0) - a_i(v_{-i}(t_0))\right) \cdot v_i(t_0) \tag{33}$$

$$\leq 0, \tag{34}$$

with the last inequality following because $\hat{a}_i(t) \leq a_i(v_{-i}(t))$ and $v_i(t) > 0$ for all $i, t \in [t_{\min}, t_{\max}]$. Having proven both $\frac{dw_i(t_0)}{dt} > 0$ and $\frac{dw_i(t_0)}{dt} \leq 0$, we conclude that no such $t'$ can exist. The other direction follows by analogous argument. $\square$

We make use of this lemma repeatedly and its application is clear so we invoke it without direct reference. We are now ready to prove the main results:

### G.1 PROOF OF THEOREM 3.1

*Proof.* At initialization, we have $\|\beta\| \geq \frac{d_1}{\sqrt{d_1+d_2}} \implies \|\beta\|\epsilon(0) \geq \frac{d_1}{d_1+d_2} = c(0)(\epsilon(0)^2 + \delta(0)^2)$. Therefore, we can remove these terms from $\frac{dc}{dt}$ at time $t = 0$, noting simple that $\frac{dc}{dt} \geq -\alpha oc$. Further, so long as $c$ is still decreasing (and therefore less than $c(0) = 1$),

$$\frac{d(\|\beta\|\epsilon - c(\epsilon^2 + \delta^2))}{dt} \geq \frac{d(\|\beta\|\epsilon - (\epsilon^2 + \delta^2))}{dt} \tag{35}$$

$$= (\|\beta\| - 2\epsilon)\frac{d\epsilon}{dt} - 2\delta\frac{d\delta}{dt} \tag{36}$$

$$= (\|\beta\| - 2\epsilon)(-c^2\epsilon + \|\beta\|c) - 2\delta(-c^2\delta) \tag{37}$$

$$= -c^2(\epsilon\|\beta\| - 2(\epsilon^2 + \delta^2)) + c(\|\beta\|^2 - 2\epsilon) \tag{38}$$

$$\geq -c(\epsilon\|\beta\| - 2(\epsilon^2 + \delta^2)) + c(\|\beta\|^2 - 2\epsilon) \tag{39}$$

$$= c(\|\beta\|^2 - 2\epsilon - \epsilon\|\beta\| + 2(\epsilon^2 + \delta^2)) \tag{40}$$

$$\geq c(\|\beta\|^2 - \epsilon(2 + \|\beta\|)). \tag{41}$$

Since $c > 0$ at all times, this is non-negative so long as the term in parentheses is non-negative, which holds so long as $\epsilon \leq \frac{\|\beta\|^2}{\|\beta\|+2}$. Further, since $\epsilon c \leq \|\beta\|$ we have

$$\frac{d\epsilon^2}{dt} = 2\epsilon\frac{d\epsilon}{dt} \tag{42}$$

$$= -2c^2\epsilon^2 + 2\epsilon c\|\beta\| \tag{43}$$

$$\leq 2\|\beta\|^2. \tag{44}$$

This implies $\epsilon(t)^2 \leq \epsilon(0)^2 + 2t\|\beta\|^2$. Therefore, for $t \leq \frac{\ln\|\beta\|/2}{2\|\beta\|}$ we have $\epsilon(t)^2 \leq \frac{1}{d_1+d_2} + \|\beta\|\ln\|\beta\|/2 \leq \frac{\|\beta\|^4}{(\|\beta\|+2)^2}$ (this inequality holds for $\|\beta\| \geq 2$). This satisfies the desired upper bound.

Thus the term in Eq. (41) is non-negative for all $t \leq \frac{\ln\|\beta\|/2}{2\|\beta\|}$, and so we have $\frac{dc}{dt} \geq -\alpha oc$ under the above conditions. Since the derivative of $o$ is negative in $c$, a lower bound on $\frac{dc}{dt}$ gives us an upper bound on $\frac{do}{dt}$, which in turn maintains a valid lower bound on $\frac{dc}{dt}$ This allows us to solve for just the ODE given by

$$\frac{dc^2}{dt} = -2\alpha c^2 o, \tag{45}$$

$$\frac{do}{dt} = -2\alpha c^2 o. \tag{46}$$

Define $m := \frac{d_1}{d_1+d_2} = \frac{1}{1+k}$. Recalling the initial values of $c^2, o$, The solution to this system is given by

$$c(t)^2 = \frac{m}{1 - \frac{(1-m)}{\exp(2\alpha mt)}}, \tag{47}$$

$$o(t) = \frac{m}{\frac{\exp(2\alpha mt)}{1-m} - 1} \tag{48}$$

$$= \frac{m}{\exp(2\alpha mt)(1 + k^{-1}) - 1} \tag{49}$$

Since these are bounds on the original problem, we have $c(t)^2 \geq m$ and $o(t)$ shrinks exponentially fast in $t$. In particular, note that under the stated condition $\sqrt{\alpha} \geq \frac{\|\beta\|\ln k}{m(\ln\|\beta\|/2)}$ (recalling $k := \frac{d_2}{d_1} > 1$), we have $\frac{\ln k}{2\sqrt{\alpha}m} \leq \frac{\ln\|\beta\|/2}{2\|\beta\|}$. Therefore we can plug in this value for $t$, implying $o(t) \leq m\left(\frac{d_1}{d_2}\right)^{\sqrt{\alpha}} = mk^{-\sqrt{\alpha}}$ at some time before $t = \frac{\ln\|\beta\|/2}{2\|\beta\|}$.

Now we solve for the time at which $\frac{dc}{dt} \geq 0$. Returning to Eq. (41), we can instead suppose that $\epsilon \leq \frac{\|\beta\|^2-\gamma}{\|\beta\|+2} \implies \|\beta\|^2 - \epsilon(2 + \|\beta\|) \geq \gamma$ for some $\gamma > 0$. If this quantity was non-negative and

has had a derivative of at least $\gamma$ until time $t = \frac{\ln k}{2\sqrt{\alpha}m}$, then its value at that time must be at least $\frac{\gamma \ln k}{2\sqrt{\alpha}m}$. For $\frac{dc}{dt}$ to be non-negative, we need this to be greater than $c(t)^2\alpha o(t)$, so it suffices to have $\frac{\gamma \ln k}{2\sqrt{\alpha}m} \geq \frac{\alpha m}{\exp(2\alpha m t)(1+k^{-1})-1} \impliedby \gamma \ln k \geq \frac{2\alpha^{3/2}m^2}{\left(\frac{d_2}{d_1}\right)^{\sqrt{\alpha}}(1+k^{-1})-1} \impliedby \gamma \geq \frac{2\alpha^{3/2}m^2 k^{-\sqrt{\alpha}}}{\ln k}$. Observe that the stated lower bound on $\alpha$ directly implies this inequality.

Finally, note that $\|b\|^2 = \epsilon^2 + \delta^2$, and therefore

$$\frac{d\|b\|^2}{dt} = 2\epsilon\frac{d\epsilon}{dt} + 2\delta\frac{d\delta}{dt} \tag{50}$$

$$= -2c^2(\epsilon^2 + \delta^2) + 2c\epsilon\|\beta\|. \tag{51}$$

Since $c(0) = 1$ and $c\epsilon < \|\beta\|$, this means $\|b\|^2$ will also be decreasing at initialization. Thus we have shown that all relevant quantities will decrease towards 0 at initialization, but that by time $t = \frac{\ln k}{2\sqrt{\alpha}m}$, we will have $\frac{dc}{dt} \geq 0$. $\qquad\square$

### G.2    PROOF OF PROOF OF THEOREM 3.2

*Proof.* Recall from the previous section that we have shown that at some time $t_1 \leq \frac{\ln k}{2\sqrt{\alpha}m}$, $c(t)^2$ will be greater than $m$ and increasing, and $o(t)$ will be upper bounded by $mk^{-\sqrt{\alpha}}$. Furthermore, $\epsilon(t)^2 \leq \frac{1}{d_1+d_2} + 2t\|\beta\|^2$. To show that the sharpness reaches a particular value, we must demonstrate that $c$ grows large enough before the point $c\epsilon \approx \|\beta\|$ where this growth will rapidly slow. To do this, we study the relative growth of $c$ vs. $\epsilon$.

Recall the derivatives of these two terms:

$$\frac{dc}{dt} = -(\epsilon^2 + \delta^2 + \alpha o^2)c + \|\beta\|\epsilon, \tag{52}$$

$$\frac{d\epsilon}{dt} = -c^2\epsilon + \|\beta\|c. \tag{53}$$

Considering instead their squares,

$$\frac{dc^2}{dt} = 2c\frac{dc}{dt} \tag{54}$$

$$= -2(\epsilon^2 + \delta^2 + \alpha o^2)c^2 + 2\|\beta\|\epsilon c, \tag{55}$$

$$\frac{d\epsilon^2}{dt} = 2\epsilon\frac{d\epsilon}{dt} \tag{56}$$

$$= -2\epsilon^2 c^2 + 2\|\beta\|\epsilon c. \tag{57}$$

Since $\delta, o$ decrease monotonically, we have $\frac{dc^2}{dt} \geq -2(\epsilon^2 + \frac{d_1}{d_1+d_2} + \alpha m \left(\frac{d_1}{d_2}\right)^{\sqrt{\alpha}})c^2 + 2\|\beta\|\epsilon$. Thus if we can show that

$$\|\beta\|\epsilon c \geq (\epsilon^2 + 2(\frac{d_1}{d_1 + d_2} + \alpha m \left(\frac{d_1}{d_2}\right)^{\sqrt{\alpha}}))c^2, \tag{58}$$

we can conclude that $\frac{dc^2}{dt} \geq (\epsilon^2 c^2 + \|\beta\|\epsilon c) = \frac{1}{2}\frac{d\epsilon^2}{dt}$—that is, that $c(t)^2$ grows at least half as fast as $\epsilon(t)^2$. And since $\delta, o$ continue to decrease, this inequality will continue to hold thereafter.

Simplifying the above desired inequality, we get

$$\|\beta\|\frac{\epsilon}{c} \geq \epsilon^2 + 2m(1 + \alpha k^{-\sqrt{\alpha}}). \tag{59}$$

Noting that $\frac{\epsilon}{c} \geq 1$ and $m = \frac{d_1}{d_1+d_2} \leq \frac{1}{2}$, and recalling the upper bound on $\epsilon(t)^2$, this reduces to proving

$$\|\beta\| \geq \frac{1}{d_1 + d_2} + 2t\|\beta\|^2 + 1 + \alpha k^{-\sqrt{\alpha}}. \tag{60}$$

Since this occurs at some time $t_1 \leq \frac{\ln k}{2\sqrt{\alpha}m}$, and since $m^{-1} = 1 + k$, we get

$$\|\beta\| \geq \frac{1}{d_1 + d_2} + \frac{\|\beta\|^2(1+k)\ln k}{\sqrt{\alpha}} + 1 + \alpha k^{-\sqrt{\alpha}}. \tag{61}$$

The assumed lower bound on $\sqrt{\alpha}$ means the sum of the first three terms can be upper bounded by a small $1 + o(1)$ term (say, $9/5$) and recalling $\|\beta\| \geq 24/5$ it suffices to prove

$$\|\beta\| \geq \frac{9}{5} + \alpha k^{-\sqrt{\alpha}} \tag{62}$$

$$\Longleftarrow \alpha k^{-\sqrt{\alpha}} \leq 3. \tag{63}$$

Taking logs,

$$\frac{2\ln\sqrt{\alpha}}{\ln k} - \sqrt{\alpha} \leq \ln 3, \tag{64}$$

which is clearly satisfied for $\sqrt{\alpha} \geq 1 + k \ln k$. As argued above, this implies $\frac{dc^2}{dt} \geq \frac{1}{2}\frac{d\epsilon^2}{dt}$ by some time $t_2 \leq \frac{\ln k}{2\sqrt{\alpha}m}$.

Consider the time $t_2$ at which this first occurs, whereby $c(t_2)^2$ is growing by at least one-half the rate of $\epsilon(t_2)^2$. Here we note that we can derive an upper bound on $c$ and $\epsilon$ at this time using our lemma and the fact that

$$\frac{dc}{dt} \leq \|\beta\|\epsilon, \tag{65}$$

$$\frac{d\epsilon}{dt} \leq \|\beta\|c. \tag{66}$$

The solution to this system implies

$$c(t_2) \leq \frac{1}{2}\left(\frac{\exp(\|\beta\|t_2) - \exp(-\|\beta\|t_2)}{\sqrt{d_1 + d_2}} + \exp(\|\beta\|t_2) + 1\right) \tag{67}$$

$$\leq \frac{1}{2}\left(\exp(\|\beta\|t_2)\left(1 + \frac{1}{\sqrt{d_1 + d_2}}\right) + 1\right) \tag{68}$$

$$\leq \frac{1}{2}\left(\exp\left(\frac{\|\beta\|\ln k}{2\sqrt{\alpha}m}\right)\left(1 + \frac{1}{\sqrt{d_1 + d_2}}\right) + 1\right), \tag{69}$$

$$\epsilon(t_2) \leq \frac{1}{2}\left(\exp(\|\beta\|t_2)\left(1 + \frac{1}{\sqrt{d_1 + d_2}}\right) + \frac{1}{\sqrt{d_1 + d_2}} - 1\right) \tag{70}$$

$$\leq \frac{1}{2}\left(\exp\left(\frac{\|\beta\|\ln k}{2\sqrt{\alpha}m}\right)\left(1 + \frac{1}{\sqrt{d_1 + d_2}}\right) + \frac{1}{\sqrt{d_1 + d_2}} - 1\right) \tag{71}$$

Then for $\alpha > \left(\frac{\|\beta\|\ln\frac{d_2}{d_1}}{m(\ln\|\beta\| - \ln 2)}\right)^2$, the exponential term is upper bounded by $\frac{\sqrt{\|\beta\|}}{2}$, giving

$$c(t_2) \leq \frac{1}{2}\left(\frac{\sqrt{\|\beta\|}}{2}\left(1 + \frac{1}{\sqrt{d_1 + d_2}}\right) + 1\right) \tag{72}$$

$$\leq \frac{\sqrt{\|\beta\|}}{2}, \tag{73}$$

$$\epsilon(t_2) \leq \frac{1}{2}\left(\frac{\sqrt{\|\beta\|}}{2}\left(1 + \frac{1}{\sqrt{d_1 + d_2}}\right) + \frac{1}{\sqrt{d_1 + d_2}} - 1\right) \tag{74}$$

$$\leq \frac{\sqrt{\|\beta\|}}{2}. \tag{75}$$

We know that optimization will continue until $\epsilon^2 c^2 = \|\beta\|^2$, and also that $\frac{dc^2}{dt} \geq \frac{1}{2}\frac{d\epsilon^2}{dt}$. Since $c \leq \epsilon$, this implies that $\epsilon^2 \geq \|\beta\|$ before convergence. Suppose that starting from time $t_2$, $\epsilon^2$ grows until

time $t'$ by an additional amount $s$. Then we have

$$s = \epsilon(t')^2 - \epsilon(t_2)^2 \tag{76}$$

$$= \int_{t_2}^{t'} \frac{d\epsilon(t)^2}{dt} \tag{77}$$

$$\leq \int_{t_2}^{t'} 2\frac{dc(t)^2}{dt} \tag{78}$$

$$= 2(c(t')^2 - c(t_2)^2). \tag{79}$$

In other words, $c^2$ must have grown by at least half that amount. Since $\epsilon(t_2)^2 \leq \frac{\|\beta\|}{4}$ and therefore $\epsilon(t')^2 \leq \frac{\|\beta\|}{4} + s$, even if $c(t')^2$ is the minimum possible value of $\frac{s}{2}$ we must have at convergence $\frac{s}{2} = c^2 = \frac{\|\beta\|^2}{\epsilon^2} \geq \frac{\|\beta\|^2}{\frac{\|\beta\|}{4} + s}$. This is a quadratic in $s$ and solving tells us that we must have $s \geq \frac{5}{4}\|\beta\|$. Therefore, $c(t')^2 \geq \frac{5}{8}\|\beta\|$ is guaranteed to occur. Noting our derivation of the loss Hessian, this implies the sharpness must reach at least $\frac{5}{8}\alpha\|\beta\|$ for each dimension of $b_o$. $\qquad\square$

## H ADDITIONAL SAMPLES UNDER VARIOUS ARCHITECTURES/SEEDS

To demonstrate the robustness of our finding we train a ResNet-18, VGG-11, and a Vision Transformer for 1000 steps with full-batch GD, each with multiple random initializations. For each run, we identify the 24 training examples with the most positive and most negative change in loss from step $i$ to step $i+1$, for $i \in \{100, 250, 500, 750\}$. We then display these images along with their label (above) and the network's predicted label before and after the gradient step (below). The change in the network's predicted labels display a clear pattern, where certain training samples cause the network to associate an opposing signal with a new class, which the network then overwhelmingly predicts whenever that feature is present.

Consistent with our other experiments, we find that early opposing signals tend to be "simpler", e.g. raw colors, whereas later signals are more nuanced, such as the presence of a particular texture. We also see that the Vision Transformer seems to learn complex features earlier, and that they are less obviously aligned with human perception—this is not surprising since they process inputs in a fundamentally different manner than traditional ConvNets.

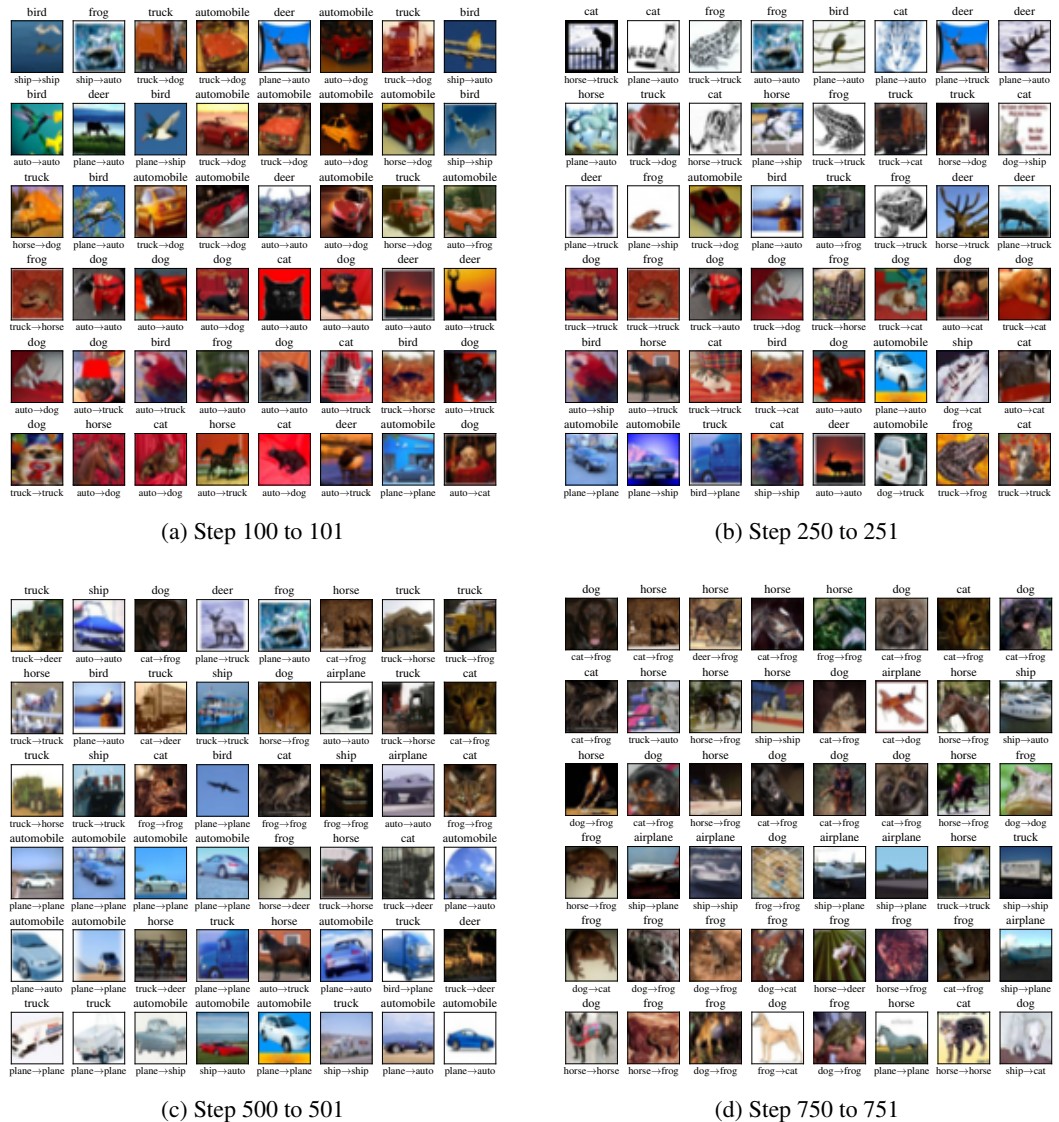

(a) Step 100 to 101

(b) Step 250 to 251

(c) Step 500 to 501

(d) Step 750 to 751

Figure 41: **(ResNet-18, seed 1)** Images with the most positive (top 3 rows) and most negative (bottom 3 rows) change to training loss after steps 100, 250, 500, and 750. Each image has the true label (above) and the predicted label before and after the gradient update (below).

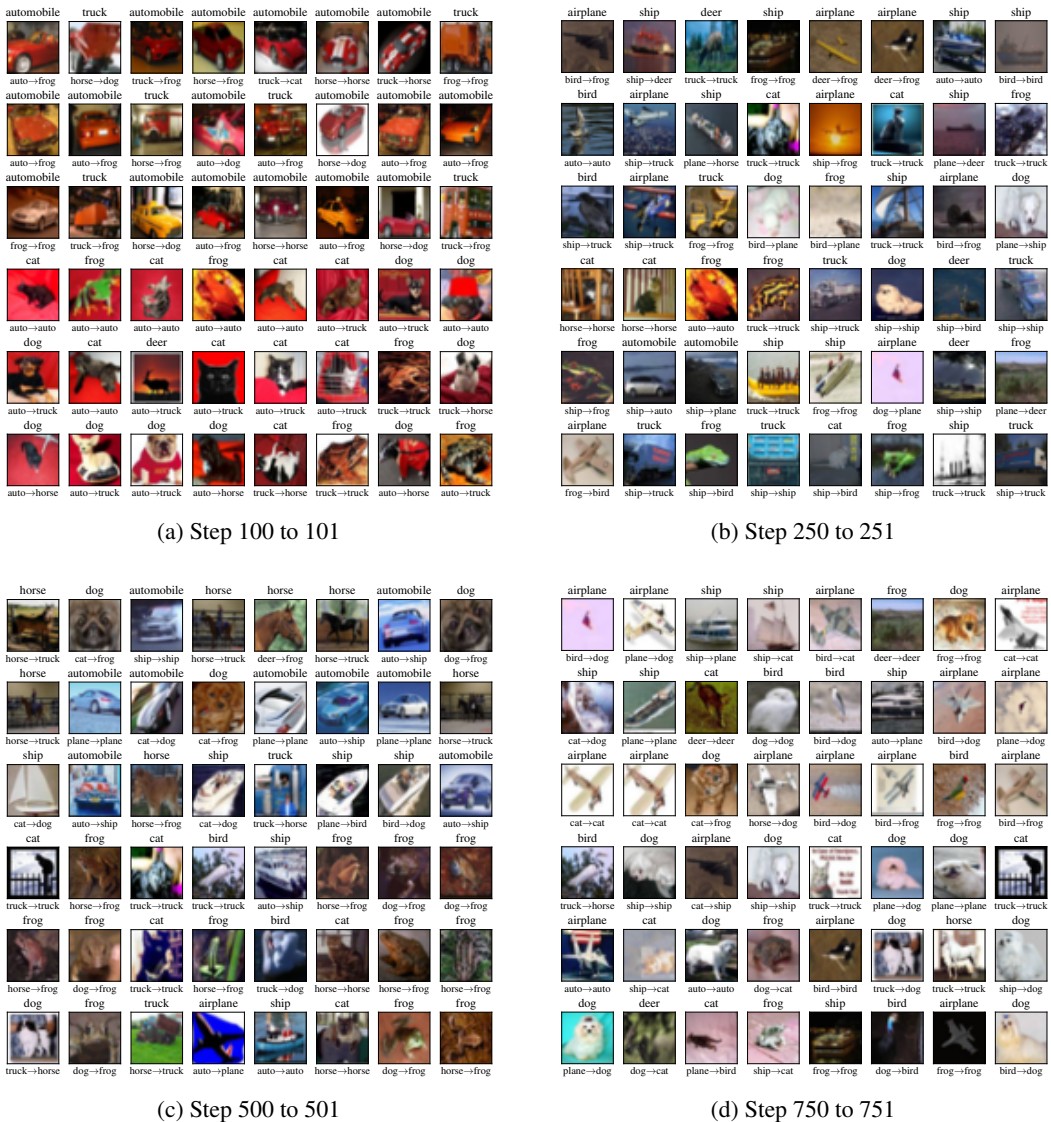

(a) Step 100 to 101

(b) Step 250 to 251

(c) Step 500 to 501

(d) Step 750 to 751

Figure 42: (**ResNet-18, seed 2**) Images with the most positive (top 3 rows) and most negative (bottom 3 rows) change to training loss after steps 100, 250, 500, and 750. Each image has the true label (above) and the predicted label before and after the gradient update (below).

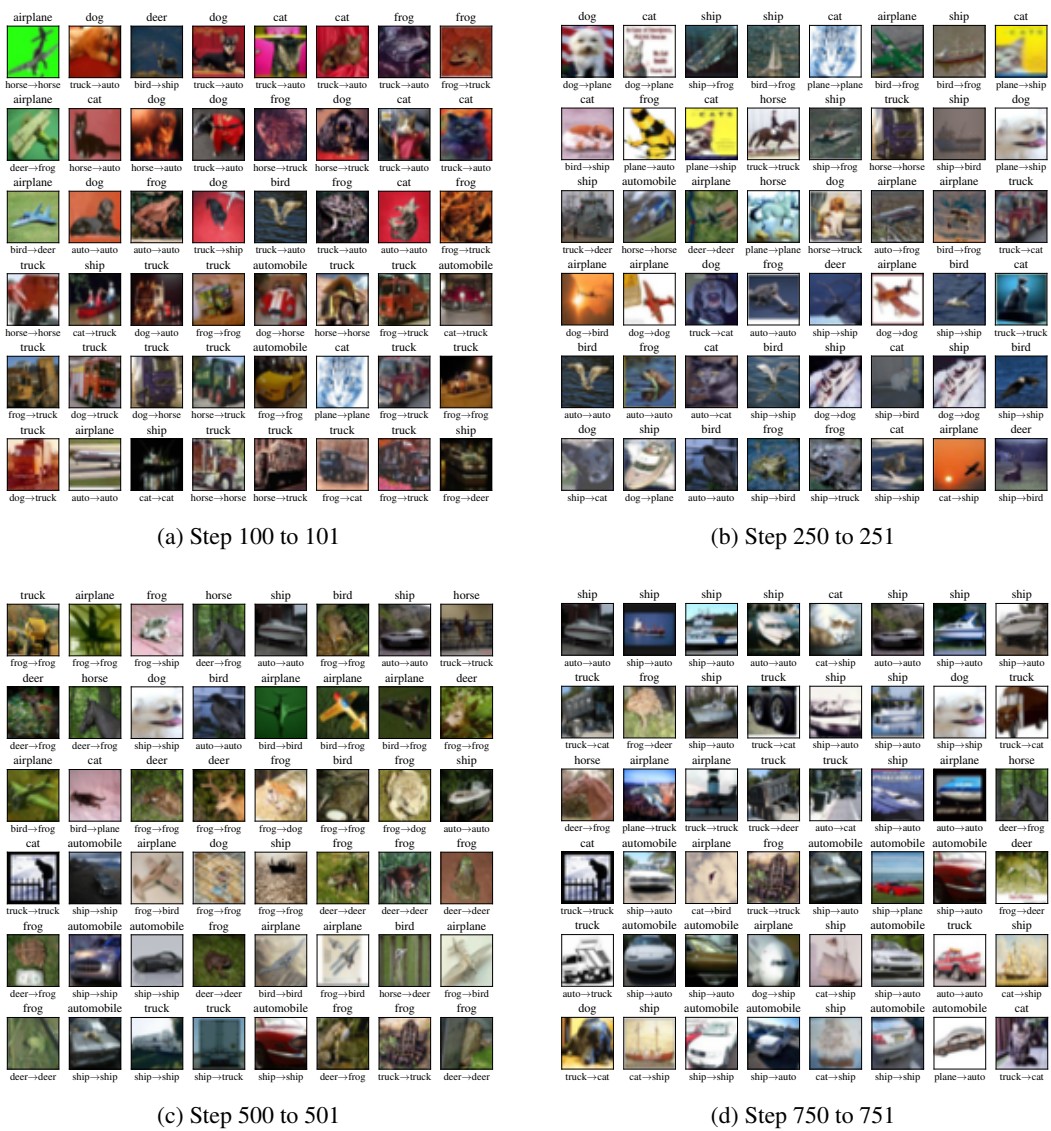

(a) Step 100 to 101

(b) Step 250 to 251

(c) Step 500 to 501

(d) Step 750 to 751

Figure 43: (**ResNet-18, seed 3**) Images with the most positive (top 3 rows) and most negative (bottom 3 rows) change to training loss after steps 100, 250, 500, and 750. Each image has the true label (above) and the predicted label before and after the gradient update (below).

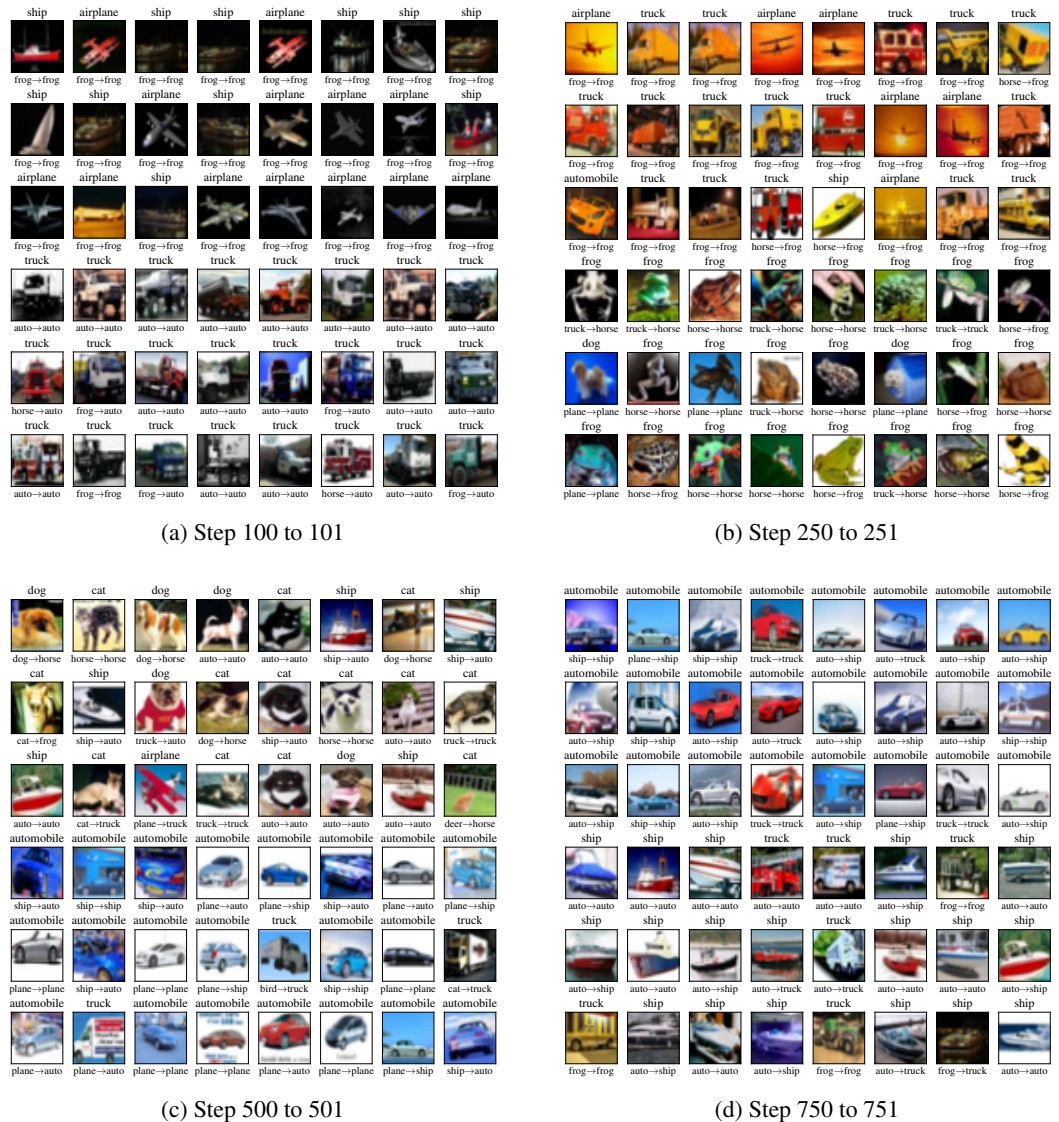

(a) Step 100 to 101

(b) Step 250 to 251

(c) Step 500 to 501

(d) Step 750 to 751

Figure 44: **(VGG-11, seed 1)** Images with the most positive (top 3 rows) and most negative (bottom 3 rows) change to training loss after steps 100, 250, 500, and 750. Each image has the true label (above) and the predicted label before and after the gradient update (below).

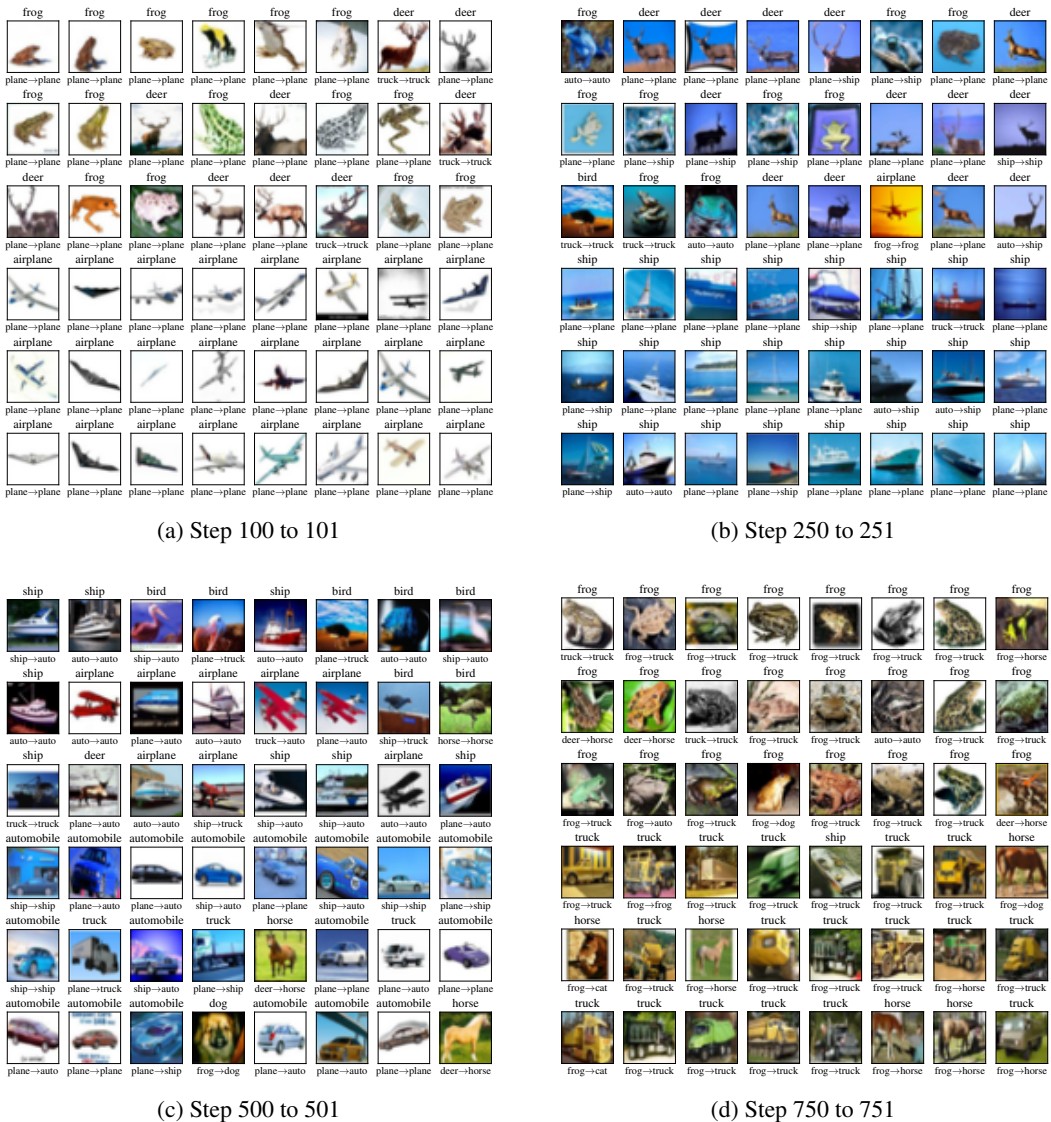

(a) Step 100 to 101

(b) Step 250 to 251

(c) Step 500 to 501

(d) Step 750 to 751

Figure 45: **(VGG-11, seed 2)** Images with the most positive (top 3 rows) and most negative (bottom 3 rows) change to training loss after steps 100, 250, 500, and 750. Each image has the true label (above) and the predicted label before and after the gradient update (below).

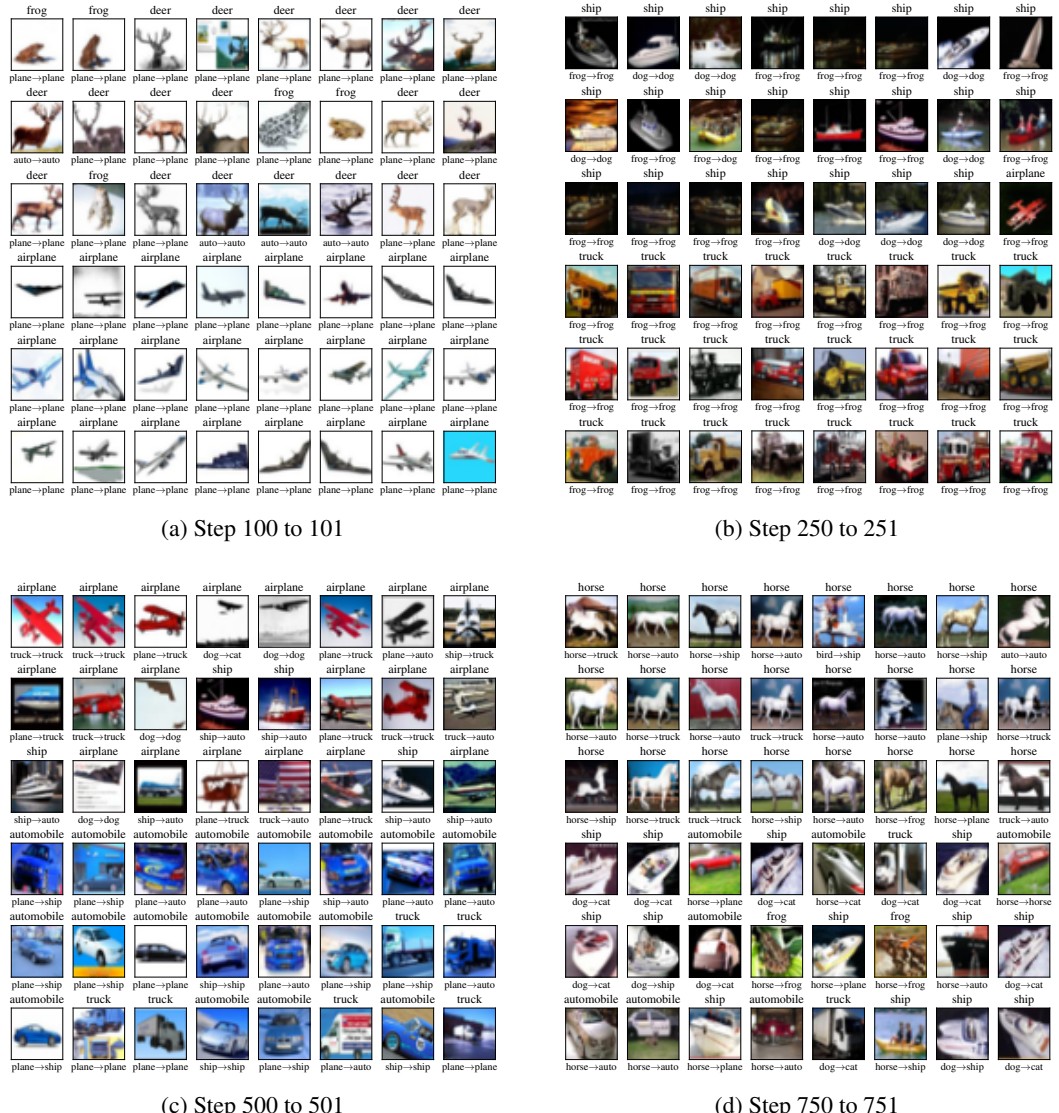

(a) Step 100 to 101

(b) Step 250 to 251

(c) Step 500 to 501

(d) Step 750 to 751

Figure 46: **(VGG-11, seed 3)** Images with the most positive (top 3 rows) and most negative (bottom 3 rows) change to training loss after steps 100, 250, 500, and 750. Each image has the true label (above) and the predicted label before and after the gradient update (below).

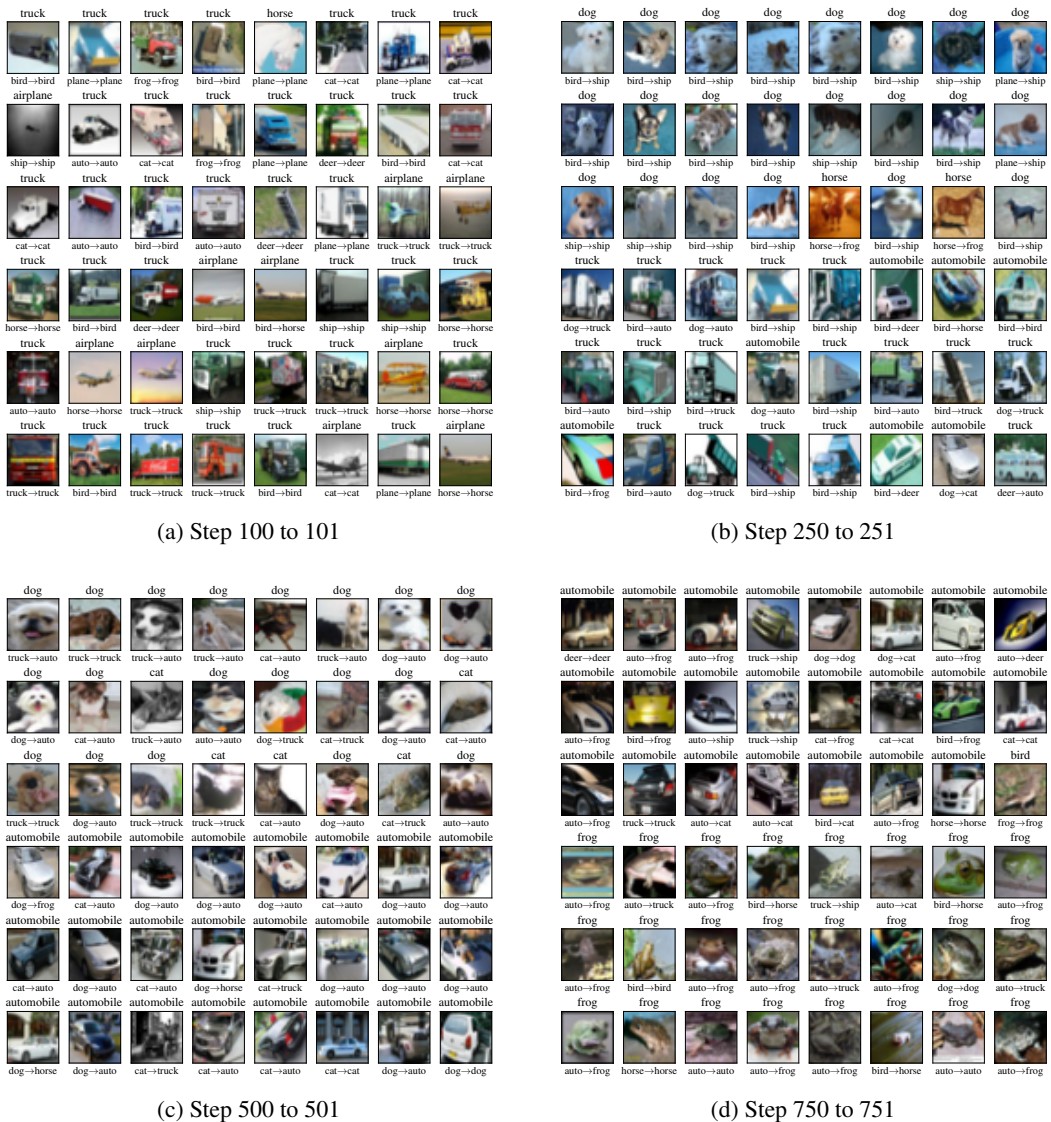

(a) Step 100 to 101

(b) Step 250 to 251

(c) Step 500 to 501

(d) Step 750 to 751

Figure 47: **(ViT, seed 1)** Images with the most positive (top 3 rows) and most negative (bottom 3 rows) change to training loss after steps 100, 250, 500, and 750. Each image has the true label (above) and the predicted label before and after the gradient update (below).

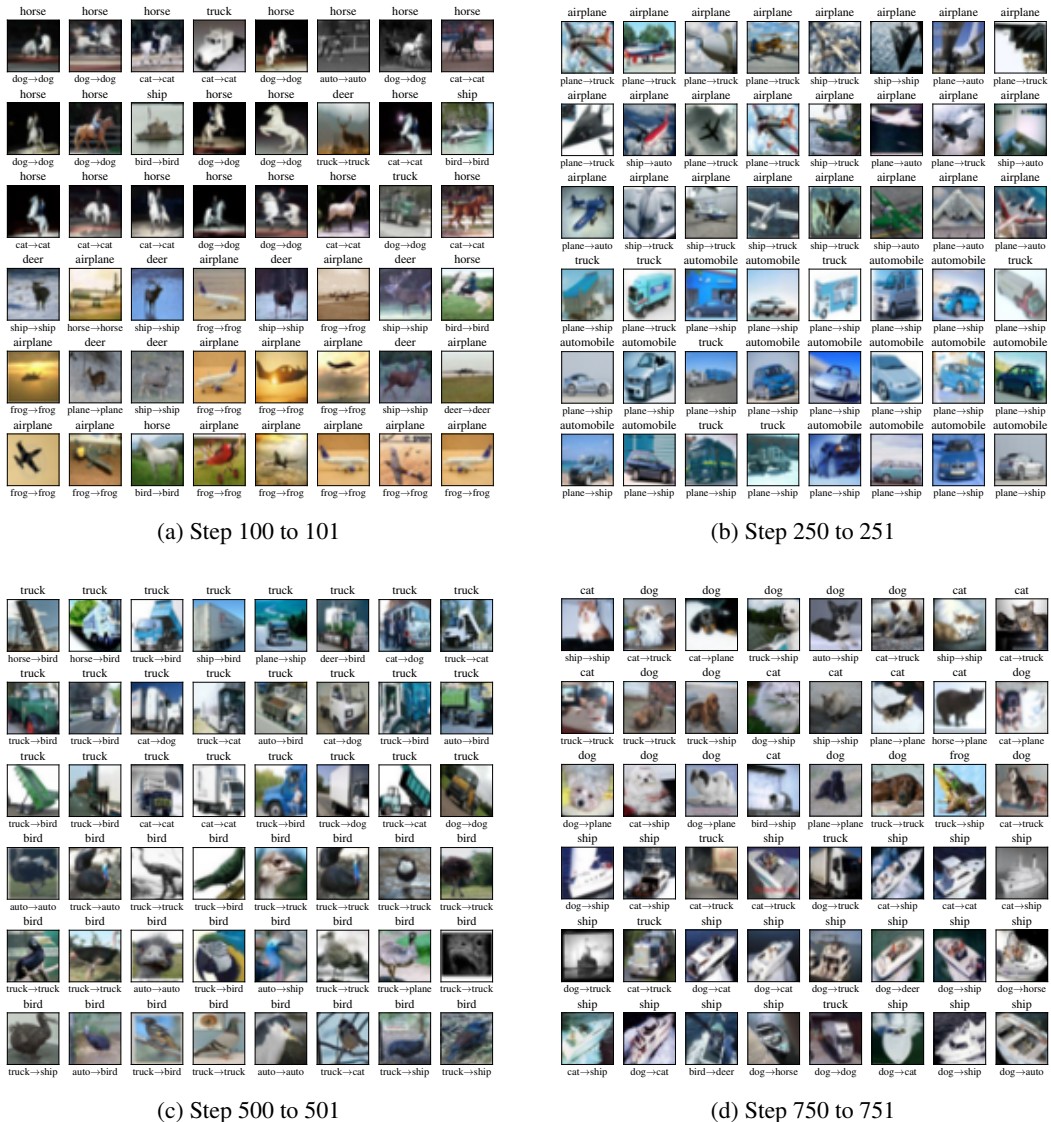

(a) Step 100 to 101

(b) Step 250 to 251

(c) Step 500 to 501

(d) Step 750 to 751

Figure 48: **(ViT, seed 2)** Images with the most positive (top 3 rows) and most negative (bottom 3 rows) change to training loss after steps 100, 250, 500, and 750. Each image has the true label (above) and the predicted label before and after the gradient update (below).

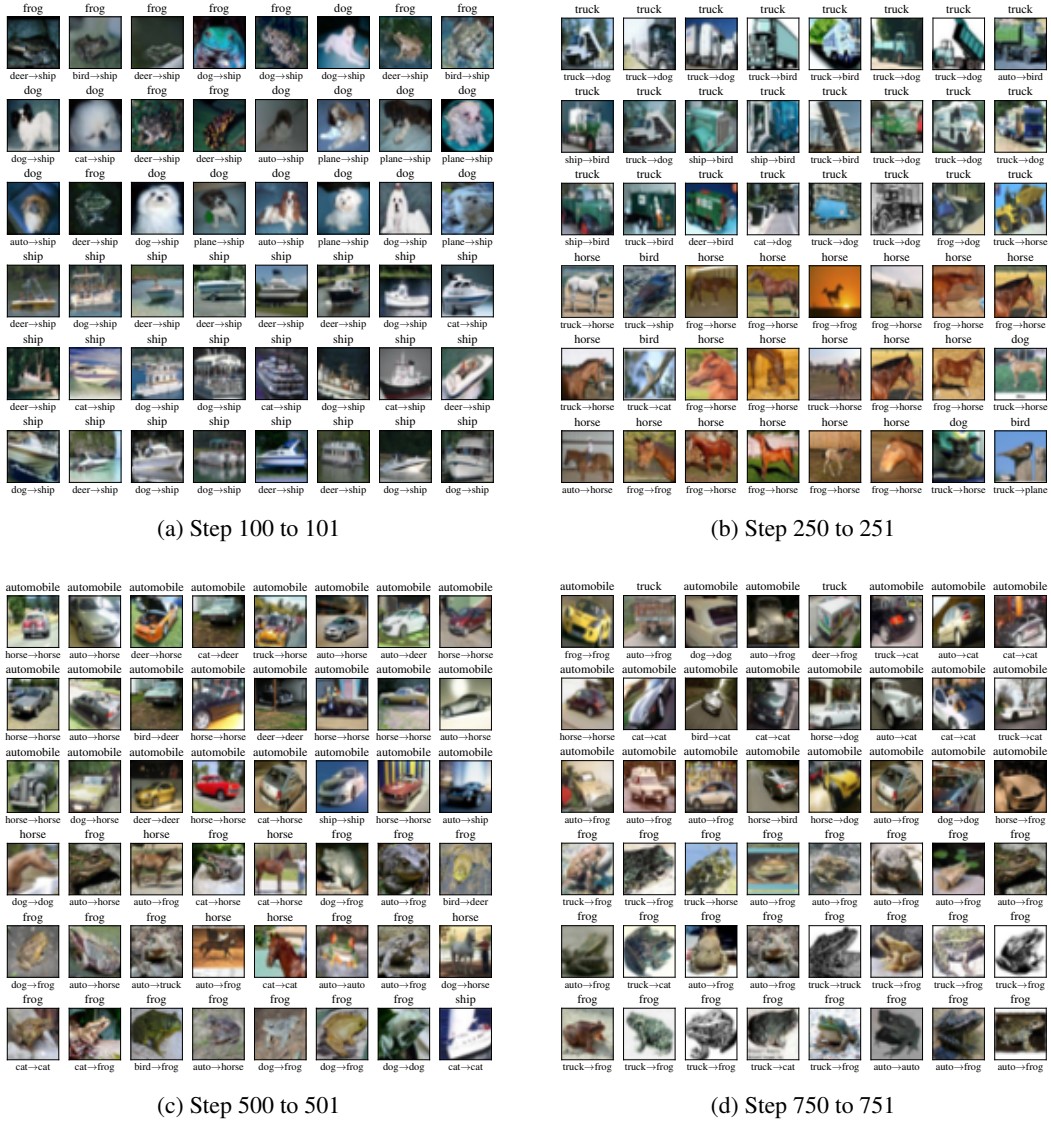

(a) Step 100 to 101

(b) Step 250 to 251

(c) Step 500 to 501

(d) Step 750 to 751

Figure 49: **(ViT, seed 3)** Images with the most positive (top 3 rows) and most negative (bottom 3 rows) change to training loss after steps 100, 250, 500, and 750. Each image has the true label (above) and the predicted label before and after the gradient update (below).

