# OpenReview forum: "Outliers with Opposing Signals Have an Outsized Effect on Neural Network Optimization"
_ICLR.cc/2024/Conference — ICLR 2024 poster_

### Official Review · Reviewer_scXv · 2023-10-27

**Soundness:** 3 good
**Presentation:** 3 good
**Contribution:** 3 good
**Rating:** 8
**Confidence:** 4

**Summary:**

This paper identifies a new phenomenon in neural network training, where some training examples exhibit opposite effects in training dynamics. Such opposite effects are signals that guide the network to learn specific features. These opposite signals influence the training dynamics significantly, sometimes producing loss spikes. Such opposite signals lead to sharpening, which happens in early layers.

There are numerous consequences of the opposite effects: (1) it allows neural networks to pick up finer features progressively (1) Batch normalization can stabilize the training dynamics (2) Adam and gradient clipping helps to mitigate the negative effects of training

**Strengths:**

I think this paper presents an interesting (though potentially controversial) method of studying neural networks. Overall it contains a consistent and plausible story of what optimizing neural networks is doing internally, thus providing a mechanistic explanation to many empirical phenomena. While I find that many notions and claims are vague, I tend to believe that there are many valuable insights based on the extensive experiments and examinations of the literature.

Here are some key strengths:
1. The phenomenon of opposing signals is plausible and seems to be a fundamental cause of many existing techniques and artifacts about neural nets.
2. The paper presents some heuristics that connect optimization with generalization, which has the potential to inspire new research
3. There are various types of experiments, ranging from simple MLPs to ResNets and Transformers.

**Weaknesses:**

An obvious weakness is the vagueness that persists in the entire paper, but this seems to be a deliberate choice by the authors. As a consequence, I find many claims confusing to understand precisely, but overall this paper gives me a better understanding of training dynamics. So I would not object to this style as much as some readers do using a scientifically more rigorous criterion. Despite the weaknesses, I am inclined to give a favorable score.

Some other weaknesses:
1.  The writing can be organized in a better way. There are many claims that are buried in long paragraphs. I would suggest that the authors number and list their claims/hypothesis/conjectures clearly, and then provide explanations for each of the claims. Even if they don't have conclusive evidence, a clear statement should be presented clearly so that future work can cite the statement unambiguously.
2. Currently there are no clear quantitative measurements. I do suggest that the authors provide one or two measurements to quantify "opposing signals" so that later work can examine their claims more precisely.

**Questions:**

1. Figure 1 suggests that at different phases of training, neural nets pick up very different features: initially features that rely on global information, and later features that rely on localized or detailed information. What is the reason for this? Do we believe that "global information" has stronger signal strength so it is learned first?
2. With opposing signals, do neural networks produce differentiated neurons, each learning a different signal? Perhaps some examination of learned neurons/representations will tell.
3. The authors mentioned 'In fact, in many cases, these features perfectly encapsulate the classic statistical conundrum of “correlation vs. causation”'. If a feature is irrelevant to a target task, why "do meaningfully correlate with it"? Why does a neural net learn features that are irrelevant to a classification task?
4. The authors mentioned that sharpness is found in the early MLP layers and the embedding layer. What about the self-attention components (considering that they are the most important part of transformers)?

---

> ### Author Response · Authors · 2023-11-18
>
> Thank you very much for your thoughtful comments and feedback!
>
> We agree that it would be ideal to more precisely define an “opposing signal” and to be able to make quantitative claims beyond our theoretical analysis in Section 3.3. Unfortunately, it is unclear which of the many possible definitions would be appropriate—both because our understanding of this phenomenon is so new, and because whether or not an image contains a particular “feature” or is an “outlier” is inherently fuzzy. We do give precise definitions for our theoretical model, but it would be difficult to define an exact measure of “skyness” in a real-world image. **To make this more clear for future readers, we have added a paragraph to Section 2 (highlighted in blue) which explicitly discusses this point.** Please let us know if you have any suggestions for how this discussion could be improved!
>
> To address your other two points in the main review:
>
> > I would suggest that the authors number and list their claims/hypothesis/conjectures clearly, and then provide explanations for each of the claims.
>
> Thanks for this suggestion: **we agree that this would improve clarity, and we will go through to highlight individual statements.** We imagine you may have been referring primarily to Sections 3.1 and 3.2, but please let us know if you had in mind any specific claims that you felt were not sufficiently clear.
>
> > I do suggest that the authors provide one or two measurements to quantify "opposing signals"
>
> We did our best to provide quantitative measurements, such as the gradient norm/curvature for individual samples (Figure 2) and the class logits/probabilities and embedding norms of “pure” color features (Figure 4 + appendix). These metrics are with respect to training points, since we are unsure how one could quantitatively measure the more conceptual idea of a “feature” / “signal”. Additionally, reviewer u2Ni suggested that we plot the distribution of changes in loss for each point over several iterations, which we have added as Figure 31 in Appendix D.
>
> To answer your additional questions:
>
> > Do we believe that "global information" has stronger signal strength so it is learned first?
>
> Essentially, yes. It is our current best understanding that global information dominates because it is the most “pervasive”—it occurs in a very large fraction of the pixels, so it heavily influences most internal activations even at random initialization. Also, after pixel normalization, features like solid colors are likely to be very large (e.g. a solid red pixel would be something like $[2.0, -2.0, -2.0]$). If a significant fraction of the pixels in an image are a certain color, these large values will propagate throughout the network. Since the local features aren’t as pervasive and are more complex than just “large values”, they produce smaller activations and only become relevant once the global features shrink. (Per Appendix C, note that the feature norm of images representing “grass texture” don’t grow nearly as large!)
>
> > do neural networks produce differentiated neurons, each learning a different signal?
>
> This is an interesting possibility! We expect that for simpler features like color this would be difficult to determine for sure. For example, if a neuron were negatively aligned with the “green” channel of an image, then it would activate for red, blue, *and* magenta images (i.e. those with no green, so 0/255 gets normalized to a large negative value). One possible way to explore this might be to experiment with different optimizers or losses to see if the presence of stronger opposing signals correlates with a larger fraction of differentiated neurons.
>
> > Why does a neural net learn features that are irrelevant to a classification task?
>
> Thanks for pointing this out—this was a poor choice of words on our part. We did not mean to convey that the feature is literally irrelevant, but rather that a human would likely ignore this feature when making a prediction. **We have reworked this line in the updated version to read:**
>
> “These features share a non-trivial correlation with the target task, but they are often not the “correct” (e.g., human-aligned) signal.”
>
> Hopefully this version is clearer.
>
> > The authors mentioned that sharpness is found in the early MLP layers and the embedding layer. What about the self-attention components?
>
> **To visualize the entire sharpness histogram, we have updated the submission with figures for all four architectures, see Appendix E**. The self-attention components certainly have some curvature, but it is nowhere near the amount found in the embedding and MLP projection layers. Generally, curvature is much less concentrated in attention models, which we believe may contribute to the particular difficulty SGD faces compared to Adam.

---

> ### Author Response · Authors · 2023-11-22
> **Discussion period ending**
>
> Hi, thanks again for taking the time to review this submission and the helpful feedback!
>
> Since the discussion period comes to a close today, we wanted to check in to make sure that we’ve answered all of your questions appropriately.
>
> We are happy to try to address any other comments in the time remaining.

---

### Official Review · Reviewer_u2Ni · 2023-10-31

**Soundness:** 2 fair
**Presentation:** 3 good
**Contribution:** 2 fair
**Rating:** 5
**Confidence:** 2

**Summary:**

This work discovers a phenomenon throughout the gradient descent training dynamics of deep neural networks: several paired groups of outlier samples emerge in the training set, where the samples of each group in a pair have a strong opposing signal to the other pair, i.e. their corresponding gradients are large and point at opposite directions. The authors hypothesize that this observation could be the reason behind the "edge of stability" and some other phenomena in deep learning optimization.

**Strengths:**

* To my knowledge, the observation is novel and can have significant impacts on our understanding of the optimization dynamics of deep neural networks.

* Many connections are established with other observations in the recent literature.

* There are experiments on a variety of setups to ensure the observations are consistent across different settings.

**Weaknesses:**

* My main concern with the current state of the submission is that the notion of outlier training samples does not seem to be precise enough and might require further clarification. For example, the authors show that the loss of almost half of the training samples may go up during a single iteration, but this change is more significant for a smaller number of outlier samples.  It is not clear what fraction of the training samples are outliers and more generally what the distribution of the loss change over all samples is.

* The relationship between the theoretical statements and the empirical observations of the paper is not entirely clear to me. The theory seems concerned with analyzing the sharpness throughout training with *gradient flow* which monotonically decreases the training loss, while a core part of the observation is to justify the spikes in the training loss that occur due to *large step size gradient descent*.

**Questions:**

* As mentioned above, it might help the readers have a better understanding if there is a visualization of the distribution of the change in loss for training samples, highlighting the frequency of the outliers in comparison with the entire training set.

* The statement of Theorem 3.2 mentions that "sharpness will increase linearly in $\Vert \beta \Vert$ until some time $t_2$". Can an exact formula (or at least more explicit upper/lower bounds) be given for sharpness as a function of time? I believe this should be possible due to having explicit formulae for the weights during training for the linear model. Additionally, is it possible to highlight the role of the problem parameters including $k, \alpha$, and $\Vert \beta \Vert$, and their relationship to the observations for deep networks?

* The example of opposing signals in text does not show the training error of the opposing outlier groups throughout training. Is it possible to recreate Figure 1 for text data?

* As a minor comment, the sharpness is typically referred to as the "largest eigenvalue of the loss", while it could be more accurately described as the "largest eigenvalue of the Hessian of the loss".

---

> ### Author Response · Authors · 2023-11-18
> **Response 1/2**
>
> Thank you for your thoughtful comments! We are very much in agreement that our finding “can have significant impacts on our understanding of the optimization dynamics of deep neural networks”. We hope our discussion below sufficiently addresses your remaining questions.
>
> > “the notion of outlier training samples does not seem to be precise enough and might require further clarification.”
>
> We agree that having a more precise definition of an “outlier”, beyond the definition we use in our theoretical analysis in Section 3.3, would be ideal. Unfortunately, such a concept is inherently fuzzy, as our primary intent was to convey something qualitative with respect to outliers in *image* space. **In our next point we incorporate your suggestion re: outliers in loss space**, but first we want to clarify that we intentionally do not try to define an outlier in image space, since this would require committing to an incomplete definition. To make this point more explicit, we have added a subsection to Section 2 (highlighted in blue). Please let us know if you have any suggestions for how this discussion could be improved.
>
> > “It is not clear…what the distribution of the loss change over all samples is.”
>
> For outliers with respect to *change in loss*, we think your idea to consider the entire distribution is a good one! We did visualize this early in our experiments: it is consistently concentrated in a nice bell curve centered near 0. **Per your suggestion, we’ve produced a plot of this distribution for a few iterations and added it to Appendix D (Figure 31).** The mean is indeed very close to 0 (typically  around $-.01$) and the standard deviation is usually around 0.3 to 0.5, sometimes a bit more, but never more than 0.8. The fraction of samples which are more than 2 (or 3 or 4, respectively) sds above/below the mean is consistently around .055 (.015 or .006, respectively). Thus the points which are changing in loss by ~1.0 or more (those depicted in Figure 1) are ~5% or less of the data, which we believe is reasonable to consider an outlier. **Thanks for this idea, we’ll add further discussion of this plot in the main body.**
>
> > “The theory seems concerned with analyzing the sharpness throughout training with gradient flow which monotonically decreases the training loss”
>
> We think our introduction of the analysis was not as clear as it could have been, and we appreciate you pointing out this confusion. Our analysis tracks the trajectory of gradient flow from the start of training up until reaching the edge of stability. Until reaching this point, the paths of GF and GD on neural networks are quite similar, particularly for small timescales (see [1])---therefore, this analysis tracks the GD path closely for “reasonable” learning rates $\eta$. The purpose of our analysis up to this point is to show that after the initial decrease, the sharpness will monotonically increase up to *at least* $\frac{2}{\eta}$, usually much more. On real data, it is empirically well-established that gradient descent will start to oscillate when this threshold is crossed. We do not prove that this also happens on our model, but we verify it with experiments in Appendix D. Our result also shows specifically that this oscillation will occur in $b_o$, the part of the network which determines how the opposing signal is used, and this bears out in our experiments.
>
> **We have made this point more explicit in a new paragraph at the beginning of Section 3.3.** Please let us know if you have suggestions for how to make this more clear!
>
> > Can an exact formula (or at least more explicit upper/lower bounds) be given for sharpness as a function of time?
>
> Because of the complex interaction between all the relevant terms, deriving bounds as a function of time that are independent of all the other variables is quite difficult. Per the proof of Theorem 3.2, we do at least have that the sharpness is upper bounded by $\mathcal{O}\left( \alpha \exp(||\beta|| t) \right)$ (see Equation 68). In terms of other variables, we can give an upper bound of $\alpha \frac{||\beta||^2}{\epsilon^2}$ (line right after Equation 75), where $\epsilon = \frac{b^\top \beta}{||\beta||}$ is the component of $b$ in the “signal” direction. We also have a matching lower bound (to order) of $\Omega\left(\alpha \frac{||\beta||^2}{\epsilon^2} \right)$. Giving an explicit lower bound for all times $t$ is much harder: our proof works by showing that the minimum satisfies a certain equilibrium, and that the relative growth rates of certain values mean we cannot reach that point without the sharpness exceeding the stated lower bound.

---

> ### Author Response · Authors · 2023-11-18
> **Response 2/2**
>
> To highlight the role of the parameters, $||\beta||$ represents the margin by which the ground truth vector can separate the points without using the opposing signals, and therefore is a proxy for the strength of the “signal”. Likewise, $\alpha$ tunes the magnitude of the opposing signals and thus represents the strength of the “noise”. Finally, $k = \frac{d_2}{d_1}$ is a measure of how “pervasive” the opposing signal is, i.e. how much it dominates the total number of dimensions.
>
> > Is it possible to recreate Figure 1 for text data?
>
> Certainly! **We’ve added this as Figure 9 in Appendix B, right alongside the text examples.**
>
> > the sharpness is typically referred to as the "largest eigenvalue of the loss", while it could be more accurately described as the "largest eigenvalue of the Hessian of the loss".
>
> Thanks for pointing out this typo. We have corrected this to “largest eigenvalue of the loss Hessian”.
>
> **We hope this response answers your questions and that you will consider increasing your score! Please let us know if you have any other questions.**
>
> [1] Continuous vs. Discrete Optimization of Deep Neural Networks. Elkabetz and Cohen, 2021

---

> ### Author Response · Authors · 2023-11-22
> **Discussion period ending**
>
> Hi, thanks again for taking the time to review this submission and the helpful feedback!
>
> Since the discussion period comes to a close today, we wanted to check in to make sure that we’ve answered all of your questions appropriately.
>
> We are happy to try to address any other comments in the time remaining.

---

> > ### Comment · Reviewer_u2Ni · 2023-11-23
> >
> > Thank you for revising the manuscript and for your detailed response. I believe that the qualitative observation of outliers with opposing signals can have significant impacts on our understanding of the optimization dynamics of deep neural networks. However, even after reviewing the additional results, I feel like there is not enough quantitative precision and clear metrics for characterizing the outliers, what fraction of the data they take up, and how strong of opposing signals they produce in comparison with the rest of the dataset. Without such quantitative characterizations, it will be difficult for follow-up works to build on and compare with the results of the current submission. Therefore, my suggestion would be for the authors to further strengthen their work by adding a possibly more quantitative analysis of the outliers in different settings, and I keep my original score.

---

### Official Review · Reviewer_5nbF · 2023-11-06

**Soundness:** 1 poor
**Presentation:** 2 fair
**Contribution:** 2 fair
**Rating:** 3
**Confidence:** 2

**Summary:**

The paper aims to understand how optimization is affected by a "opposing signals" --- subset of signals that can be associated with different classes depending on the rest of the signals ---  from small groups. For example, skies appear in backgrounds of planes most often, but they also appear in backgrounds of boats. The main claim of the paper is that such signals affect the progress of optimization by forcing the model to approach a basin in loss landscape that balances the opposing the forces. The paper shows experimenting track instability.

**Strengths:**

The paper seems to identify an interesting phenomenon where the loss landscape that models end up in can be affected by opposing signals from small groups in data. There are some interesting plots in the paper that show case how the model depends on certain features (like the sky color).

**Weaknesses:**

There is too much informal language and the paper does not consider other confounding factors in leading to the instability. Specifically, the paper uses "per-step largest loss change" to define groups. Of course these groups will have large gradient norm because they have the largest errors in them.

1. There are words like like genuine, correct, and random noise used but without formal definition. The authors acknowledge this but without clarity on what these are, the proposed phenomenon is too vague. You can define features are the groups of data that the feature helps correctly classify,  which becomes a formal definition.

2. The synthetic experiment has a non-linear ground truth (absolute value of x_0), is that what contributes to the reduction in sharpness?

3. What about the roles of batchsize, learning rate, lr schedulers, other kinds of regularization?


One of my issues with the writing is loose language. For example, why call $x_0$ unimportant? Clearly it helps predict the output, just on a subset of data. The idea of opposing signals seems very similar to have a "shortcut". The text in section 3 seems tedious because the authors use language the seems to be intentionally removed from terminology in shortcut learning  or distribution shift, but they use an example that is commonly called shortcut learning (model rely on skies to classify planes because many planes have sky in the background).


Finally, the main claim is "opposing signals affect shaprness like so ... ". Then, there should be a single plot that shows this exactly. I could find one such experiment. Why not directly measure sharpness, at least for the first few layers? It seems like this was done in "Additional findings", but there is not plot and supporting text. It would be great if there was a plot that showed, training instability occur exactly when sharpness rises up, and then controlling opposing signals via some robust loss or remove some samples both removes the sharpness and training instability.


Overall, the idea is interesting but the experiments need to be more than a collection of simple post-hoc checks to show that the cause of optimization is certain groups. One direction, maybe is to randomly change the backgrounds of the supposed outlier groups (upto rho fraction) and show the sharpness behavior is avoided.

**Questions:**

See weaknesses.

---

> ### Author Response · Authors · 2023-11-18
> **Response 1/2**
>
> Thanks for your review. We understand that your primary concern is with regard to our use of informal language, and the fact that we do not provide formal definitions for certain terms such as “features” (beyond our theoretical analysis). In our detailed response below, we will explain our choice of language in several places you pointed out, and we are happy to make adjustments to clarify our message.
>
> ## On the use of informal language
>
>
> > There are words like like genuine, correct, and random noise used but without formal definition. The authors acknowledge this but without clarity on what these are
>
> We agree that it would be ideal to be able to give precise definitions for many of these concepts, beyond the formal definitions we give in our analysis in Section 3.3. Unfortunately, **choosing the "correct" formal definition for real data seems infeasible to us, given our nascent understanding of opposing signals:** e.g., what a “feature” is and whether or not an image contains it is inherently fuzzy (how would one measure what it means to have the “sky” feature, or quantify the “skyness” of a real-world image? Is one blue pixel enough?)
>
> You’ve suggested, for example, defining a “feature” as “the groups of data that the feature helps correctly classify”---we assume you meant here that a feature would be a *property shared by such a group*, rather than the groups themselves. But **this would still not be precise enough:** if planes have a sky background and are also consistently white, are these two separate features? Or just one? What about $x_o$ in our theoretical model (if we consider classification instead of regression), which is certainly a “feature”, but is unable to help a linear classifier separate the two classes and therefore doesn’t fit your definition? Because of this difficulty, outside of theoretical models like the one we study in Section 3.3, it is unclear which of the many possible definitions would be appropriate—especially because our understanding of this phenomenon is so new. Furthermore, we are not aware of any prior work that does this.
>
> We do want to make it as clear as possible that we *intentionally* do not attempt to define “feature” or other necessarily hazy concepts in this work. At several points in the paper (e.g., second paragraph of the intro) we point out that we are presenting our current best understanding only at a high level. **To make this more explicit, we have added a subsection to Section 2 (highlighted in blue) which discusses this point.** We hope this clarifies our intent.
>
> > There is too much informal language…the paper uses "per-step largest loss change" to define groups. Of course these groups will have large gradient norm because they have the largest errors in them.
>
> **This is not how we define the groups.** Per-step loss change is how we identify candidate samples which have significant influence—we define the groups via visual inspection (note that if you track our groups throughout training, they do not *always* have large gradient norm). We are very forthright in the first paragraph of Section 2 that these groups are not formally defined but that we feel they are self-evident. When identifying patterns in real-world data, **sometimes the best you can do is to say “we think this is clear, here are some examples so you can judge for yourself”, as we do in Appendix H.** As a thought experiment: suppose instead we were studying a group of CIFAR images that we claim are mislabeled—how could we define this set, other than to visualize them and say “look for yourself”? Clearly, “images whose true label doesn’t match the label in the dataset” would not be a valid definition, since the “true label” cannot be formally defined.

---

> ### Author Response · Authors · 2023-11-18
> **Response 2/2**
>
> ## To address your remaining comments:
>
> > why call $x_o$ unimportant?
>
> **We believe you may have misread this sentence.** We call the *choice of model distribution* over $x_o$ unimportant, not $x_o$ itself. That is, we are saying that our theoretical results will hold even if it follows a different distribution. We have updated this sentence to make this distinction, please let us know if it is unclear. You are correct that “it helps predict the output, just on a subset of data”, but the problem is that we cannot know a priori which subset a sample belongs to, which means the optimal thing to minimize loss is to not use $x_o$ at all.
>
> > The idea of opposing signals seems very similar to have a "shortcut".
>
> We certainly agree that there is a clear connection between “shortcuts” and opposing signals, and **we point this out in the discussion in Appendix A** (opposing signals are related to many other concepts so the full discussion would not fit in the main body). You are also correct that our running example of “sky” could fit the traditional idea of a shortcut. However, we need to use more generic terminology in Section 3 because **opposing signals and “shortcuts” are not the same.**
>
> As one example, “shortcuts” refer to correlations in the data which allow a classifier to “cheat” by not learning the true decision rule. But one of the opposing signals in Figure 1 is the presence of vehicular features (we can’t define this exactly, but the images all share things like wheels, headlights, windshields, etc.). **These features are in fact the *intended* signal for classifying cars and trucks, so they would not be considered shortcuts.** Furthermore, we describe in Section 3.1 how opposing signals are specifically a way to reduce loss as quickly as possible according to the *current* internal representation, in contrast to “shortcuts” which are fixed statistical relationships of the data which some trained neural networks rely on. So while we agree that there is a clear connection—which we do discuss—there are key differences and it is important to describe opposing signals more generally.
>
> > Why not directly measure sharpness, at least for the first few layers? It seems like this was done in "Additional findings", but there is not plot and supporting text.
>
> We agree that it would be good to include the figures showing the result we reported in the additional findings. **We have added these figures for all architectures to the updated submission, please see Appendix E.**
>
> > It would be great if there was a plot that showed, training instability occur exactly when sharpness rises up
>
> This is already known to occur. When the sharpness crosses the threshold $\frac{2}{\eta}$, Gradient Descent is no longer guaranteed to reduce the local quadratic Taylor expansion, and instability occurs. See [1] for examples of this and further discussion.
>
> > and then controlling opposing signals via some robust loss or remove some samples both removes the sharpness and training instability.
>
> Figure 2 demonstrates that these individual samples have substantially higher curvature along the top eigenvector than the average point. This *necessarily implies* that removing them would reduce the sharpness. For a similar result regarding robust loss, in Section 3.4 we point to [2], who already show that label smoothing reduces sharpening.
>
> > One direction, maybe is to randomly change the backgrounds of the supposed outlier groups (upto rho fraction) and show the sharpness behavior is avoided.
>
> We are unsure how one could reliably run this experiment. The images are in raw pixel form—how would we “randomly change the backgrounds”? Also, please keep in mind that **opposing signals are not *only* in the image background; we chose this only as a simple running example** (e.g., in Figure 1 two of the signals are “vehicular features” and “grass texture”). This means that changing the background would not necessarily reduce sharpness.
>
> **We hope this response answers all your questions and makes clear why we did not try to formally define some of these inherently fuzzy concepts. We’ve added Subsection 2.1 to try to make this more explicit. Please let us know if you have any other questions.**
>
> [1] Gradient Descent on Neural Networks Typically Occurs at the Edge of Stability. Cohen et al. 2021
>
> [2] On progressive sharpening, flat minima and generalisation. MacDonald et al. 2023

---

> ### Author Response · Authors · 2023-11-22
> **Discussion period ending**
>
> Hi, thanks again for taking the time to review this submission and the helpful feedback!
>
> Since the discussion period comes to a close today, we wanted to check in to make sure that we’ve answered all of your questions appropriately.
>
> We are happy to try to address any other comments in the time remaining.

---

### Official Review · Reviewer_VsCH · 2023-11-06

**Soundness:** 3 good
**Presentation:** 3 good
**Contribution:** 3 good
**Rating:** 6
**Confidence:** 3

**Summary:**

This paper studies the influence of samples w/ large, opposing features, and provides explanations for several prior observations.

**Strengths:**

This paper explores the Outliers with Opposing Signals on the training process of neural networks.
The authors support their findings with extensive experiments and bolster their claims through the use of toy models.
As applications, the authors also provide new understanding of Progressive Sharpning and EoS.

**Weaknesses:**

Please see "Questions".

**Questions:**

1. In EoS, vibrations are observed along the maximal eigen-direction. Could the authors explain how the presence of outliers impacts this phenomenon?

2. In [1], the authors proposed the concept of multi-view features, which seems to be similar to the Outliers with Opposing Signals in this work. Could the author provide some comparisons?

[1] Allen-Zhu and Li. Towards understanding ensemble, knowledge distillation and self-distillation in deep learning. (ICLR 2023)

---

> ### Author Response · Authors · 2023-11-18
>
> Thanks for your review! We are happy to answer your questions:
>
> > In EoS, vibrations are observed along the maximal eigen-direction. Could the authors explain how the presence of outliers impacts this phenomenon?
>
> Our experiments suggest that the oscillations along the top eigenvector are *caused* by opposing signals. This maximal eigenvector represents **both** (i) the direction with a dominant gradient (see Figure 2; this is because opposing signals have large feature norm); **and** (ii) the direction that will most rapidly imbalance the groups, and thus it has high loss curvature. Item (i) implies that the overall loss gradient is heavily aligned with this top eigenvector—at the same time, because of the curvature described in Item (ii), we will overshoot the minimum, and so the next gradient will point back in the other direction. These two components together cause the phenomenon you’ve noted (we plot this for SGD in Figure 6).
>
> We believe Figure 4 presents convincing evidence of this: during training, the class logits on a sky-colored image bounce back and forth between the “plane” class and a more uniform distribution. As shown in the rightmost subfigure, the feature norm is very large, which causes high curvature [Item (ii)], so the parameters keep overshooting the minimum. And the first two subfigures depict how this oscillation direction (i.e., the gradient we are following) is aligned with the sky feature [Item (i)], causing large changes to the class probabilities.
>
> > In [1], the authors proposed the concept of multi-view features, which seems to be similar to the Outliers with Opposing Signals in this work. Could the author provide some comparisons?
>
> Thank you for bringing this work to our attention! We expect there are many possible properties of a dataset that could give rise to opposing signals, and multi-view features are one plausible candidate. For example, one could imagine that the multi-view structure could determine probabilities of observing certain combinations of features, which in turn would decide how many samples exhibit a particular feature in each group and how large the gradient is in each of the two directions. Our observations are also consistent with that work’s result about how ensembling improves performance by reducing individual networks’ reliance on particular features, for example if each model learns to downweight opposing signals to different degrees, but no single model eliminates all of them. More fine-grained experimentation on this idea would be an interesting direction for future work.
>
> As we noted in the paper, we believe in many cases our experiments provide strong evidence *in support of* the models in prior work (such as multi-view features), implying that these models realistically capture optimization in deep learning. **We will add this discussion to the related work.**
>
> **We hope this response answers your questions and that you will consider increasing your score! Please let us know if you have any other questions.**

---

> ### Author Response · Authors · 2023-11-22
> **Discussion period ending**
>
> Hi, thanks again for taking the time to review this submission and the helpful feedback!
>
> Since the discussion period comes to a close today, we wanted to check in to make sure that we’ve answered all of your questions appropriately.
>
> We are happy to try to address any other comments in the time remaining.

---

### Meta-Review · Area_Chair_VZqN · 2023-12-03

**Metareview:**

The paper presents an insightful analysis of the phenomenon that training of deep neural networks reaches the edge of stability/break-even point in which it's high unstable. The main idea is that the increase in sharpness (that results in high instability) can be attributed to a small group of examples that have opposed gradients due to using the same highly predictive features. The presented angle is novel and might transfer into novel optimization methods in the future.

The main concern of Reviewers was the imprecise language that is used by the Authors. Relatedly, the Authors decided no to try to precisely define the phenomenon (e.g. the precise algorithm for how to identify outliers is missing). The theoretical model helps to clarify the intuition, but does not fully alleviate this issue. This shortcoming is however alleviated by the clarity in writing that helps the reader understand the meaning behind somewhat informal language. The reviewers have not engaged in further discussion with the Authors.

Despite the drawbacks, the paper —as noted by one reviewer— can have a significant impact on our understanding of the optimization dynamics of deep neural networks. All in all, I believe that the findings are novel and interesting enough to warrant acceptance despite certain shortcomings of the paper. I would be very grateful if the Authors could consider making the following changes to the manuscript: (a) making more precise how to find outliers for others and (b) making a clearer connection to the field of shortcut learning. Please remember to address other comments by the reviewers.

**Justification For Why Not Higher Score:**

The paper would be improved if it used more precise language (which doesn't have to take away from the intuitive exposition)

**Justification For Why Not Lower Score:**

The paper holds promise to offer a significantly better understanding of an important learning dynamics of neural networks that seems to be exhibited by vast majority of neural networks.

---

### Decision · Program_Chairs · 2024-01-16

Accept (poster)